# A monoacylglycerol lipase inhibitor showing therapeutic efficacy in mice without central side effects or dependence

Ming Jiang[1,9], Mirjam C. W. Huizenga[1,9], Jonah L. Wirt[2,9], Janos Paloczi[3], Avand Amedi[1], Richard J. B. H. N. van den Berg[4], Joerg Benz[5], Ludovic Collin[5], Hui Deng[1], Xinyu Di[6], Wouter F. Driever[1], Bogdan I. Florea[4], Uwe Grether[5], Antonius P. A. Janssen[1], Thomas Hankemeier[6], Laura H. Heitman[7], Tsang-Wai Lam[8], Florian Mohr[1], Anto Pavlovic[5], Iris Ruf[5], Helma van den Hurk[8], Anna F. Stevens[1], Daan van der Vliet[1], Tom van der Wel[1], Matthias B. Wittwer[5], Constant A. A. van Boeckel[1], Pal Pacher[3], Andrea G. Hohmann[2] ✉ & Mario van der Stelt[1] ✉

Monoacylglycerol lipase (MAGL) regulates endocannabinoid 2-arachidonoylglycerol (2-AG) and eicosanoid signalling. MAGL inhibition provides therapeutic opportunities but clinical potential is limited by central nervous system (CNS)-mediated side effects. Here, we report the discovery of LEI-515, a peripherally restricted, reversible MAGL inhibitor, using high throughput screening and a medicinal chemistry programme. LEI-515 increased 2-AG levels in peripheral organs, but not mouse brain. LEI-515 attenuated liver necrosis, oxidative stress and inflammation in a $CCl_4$-induced acute liver injury model. LEI-515 suppressed chemotherapy-induced neuropathic nociception in mice without inducing cardinal signs of $CB_1$ activation. Antinociceptive efficacy of LEI-515 was blocked by $CB_2$, but not $CB_1$, antagonists. The $CB_1$ antagonist rimonabant precipitated signs of physical dependence in mice treated chronically with a global MAGL inhibitor (JZL184), and an orthosteric cannabinoid agonist (WIN55,212-2), but not with LEI-515. Our data support targeting peripheral MAGL as a promising therapeutic strategy for developing safe and effective anti-inflammatory and analgesic agents.

The cellular process of lipid metabolism is intricately controlled and encompasses enzymes that govern the spatial and temporal aspects of lipid biosynthesis and degradation. Lipid homoeostasis is frequently disrupted in diverse conditions, including inflammation, pain and cancer[1–3]. Monoacylglycerol lipase (MAGL) is a serine hydrolase located at the membrane, tasked with the hydrolysis of monoacylglycerols into free fatty acids (FFAs) and glycerol. This enzyme utilizes a catalytic triad composed of serine, histidine and aspartate to cleave the ester bond in monoacylglycerols, but not in di- or triacylglycerols as substrates[4,5]. MAGL is the principal enzyme responsible

[1]Department of Molecular Physiology, Leiden University & Oncode Institute, Leiden, Netherlands. [2]Department of Psychological and Brain Sciences, Program in Neuroscience, Gill Center for Biomolecular Science, Indiana University, Bloomington, IN, USA. [3]Laboratory of Cardiovascular Physiology and Tissue Injury, National Institute of Health/NIAAA, Rockville, MD, USA. [4]Department of Bio-organic Synthesis, Leiden University, Leiden, Netherlands. [5]Roche Innovation Center Basel, F. Hoffmann-La Roche Ltd., Basel, Switzerland. [6]Metabolomics and analytics center, Leiden University, Leiden, Netherlands. [7]Division of Drug Discovery and Safety, Leiden University & Oncode Institute, Leiden, Netherlands. [8]Pivot Park Screening Centre, Oss, Netherlands. [9]These authors contributed equally: Ming Jiang, Mirjam C. W. Huizenga, Jonah L. Wirt. ✉e-mail: hohmanna@indiana.edu; m.van.der.stelt@chem.leidenuniv.nl

for the degradation of the endocannabinoid 2-arachidonoylglycerol (2-AG) in the brain[6]. 2-AG serves as an endogenous ligand for cannabinoid $CB_1$ and $CB_2$ receptors, playing a crucial role in diverse physiological processes, such as appetite regulation, pain perception, emotional responses, and energy homeostasis[7]. The hydrolysis of 2-AG by MAGL terminates the activation of CB receptors, releasing free arachidonic acid (AA). Subsequently, AA serves as a precursor for the synthesis of pro-inflammatory eicosanoids, including prostaglandins[3,8].

MAGL has emerged as a promising therapeutic target for a range of disorders due to its pivotal involvement in endocannabinoid and eicosanoid signalling. This includes (neuro)inflammatory and neuropsychiatric diseases, as well as conditions involving acute tissue injury and cancer[9–12]. CNS penetrant MAGL inhibitors were active in various preclinical models of neuropathic pain, including models of traumatic nerve injury and chemotherapy-induced peripheral neuropathy, via cannabinoid $CB_1$ and $CB_2$ receptor mechanisms[13,14]. Chronic exposure to the irreversible inhibitor JZL184 resulted, however, in unwanted central nervous system (CNS)-mediated side effects. JZL184 induced pharmacological tolerance, physical dependence, impaired synaptic plasticity and $CB_1$ receptor desensitization in the nervous system in rodents[15]. The irreversible MAGL inhibitor Lu-Ag06466 developed by Lundbeck (former ABX-1431 from Abide Therapeutics), which is currently tested in phase 2 clinical trials in patients with focal epilepsy, post-traumatic stress disorder and spasticity in multiple sclerosis, also exerted adverse CNS effects[16]. Thus, alternative strategies that exploit the beneficial therapeutic effects of MAGL inhibition without inducing deleterious consequences are highly desirable.

The majority of documented MAGL inhibitors, such as JZL184 and ABX-1431, operate through an irreversible mechanism. They create a covalent complex after a nucleophilic attack by the catalytic serine[17–19]. While irreversible inhibitors may display increased potency and a long target residence time that can drive pharmacological efficacy, their permanent inhibition of the target uncouples the pharmacodynamic response from the pharmacokinetics properties[20]. Mechanism-based off-target activity may pose a liability of irreversible inhibitors, which necessitates a large therapeutic window to avoid the risk associated with overdosing. Moreover, irreversible inhibitors have the potential for idiosyncratic drug-related toxicity due to hapten-formation[20,21]. It is proposed that reversible inhibitors may allow for better dosing regimens and have an improved safety profile, thereby avoiding undesirable side effects. However, most of the currently described reversible MAGL inhibitors lack potency, selectivity, or appropriate physicochemical properties for in vivo studies using an oral route of administration[22–28]. Therefore, it is important to identify novel chemotypes to reversibly inhibit MAGL and to investigate their therapeutic potential.

Here, we report the discovery of LEI-515 as a peripherally restricted, reversible MAGL inhibitor. High throughput screening (HTS) and a medicinal chemistry programme guided by activity-based protein profiling led to the identification of LEI-515 as a potent MAGL inhibitor with excellent pharmacokinetic properties. A co-crystal structure revealed its binding mode in MAGL. LEI-515 increased cellular 2-AG levels and dose-dependently increased 2-AG levels in the intestines, but not in the brain. LEI-515 attenuated $CCl_4$-induced acute liver necrosis, inflammation, and oxidative stress. In addition, the compound exerted anti-nociceptive effects in a mouse model of paclitaxel-induced neuropathy without inducing cardinal signs of $CB_1$-mediated CNS side effects, tolerance, or physical dependence. Antinociceptive efficacy was preserved after repeated dosing and blocked by cannabinoid $CB_2$ receptor antagonists that differ in their ability to penetrate the CNS, but not by $CB_1$ receptor antagonists. Taken together, LEI-515 is a first-in-class peripherally restricted, reversible MAGL inhibitor that increases 2-AG levels in the periphery and reduces tissue injury, inflammation, and neuropathic pain without producing untoward effects associated with direct $CB_1$ receptor activation in the brain.

## Results

### High throughput screening and hit triaging

To identify new reversible inhibitors for MAGL, we performed an HTS campaign in collaboration with the Pivot Park Screening Centre. A previously reported fluorescent MAGL activity assay using the natural substrate 2-AG was converted to an HTS-compatible 1536-well format (Fig. 1a). As an enzyme source, we used membrane fractions of human embryonic kidney 293T (HEK293T) cells that were transiently transfected with human MAGL. This resulted in a robust assay with an average $Z' = 0.83$, signal-to-background ratio of 8.7, and inter-plate variability (coefficient of variability) of 4.0%. Approximately 234,000 compounds were screened at a single concentration (10 μM) in 188 plates over 2 days, affording 1555 actives with more than 50% inhibition ($z$-score < −4.95), yielding a hit rate of 0.67% (Fig. 1b and c). Utilizing a nearest-neighbour clustering model, potential false negatives were incorporated into the initial hit list, yielding 4389 compounds. Subsequent active confirmation was conducted using the exact assay conditions, leading to a list of confirmed actives consisting of 1142 compounds. To eliminate false positives, a deselection assay was implemented with glycerol as the substrate instead of 2-AG. Following deselection, the 334 remaining compounds were further narrowed down to 146 by applying a more stringent criterion: >60% inhibition at 10 μM in the primary assay. In the subsequent step, structures were scrutinized, and compounds displaying an apparently irreversible mode of action—determined by the presence of chemical motifs commonly found in irreversible serine hydrolase inhibitors, such as lactones or activated carbamates/ureas—were excluded. Additionally, compounds with unfavourable physicochemical properties or those protected by intellectual property were removed. This triage process resulted in a qualified hit list of 50 compounds, and the chemical structures are disclosed in (Supplementary Table 1). Within the qualified hit list, there were 9 distinct clusters featuring different chemotypes. Notably, benzoxazine derivatives and piperazine amides had been previously reported as MAGL inhibitors[28,29]. Many of the identified hit exhibited favourable physicochemical properties, such as a molecular weight (MW) below 500 g/mol and a calculated partition coefficient (cLogP) below 5. The topological Polar Surface Area (tPSA) of nearly all compounds was below 90 Å$^2$, which is generally considered the upper limit for molecules to effectively cross the blood–brain barrier. Concentration-response experiments were conducted for all hits, and half-maximal inhibitory concentration values (expressed as negative log ($pIC_{50}$)) were determined at various time points. The $pIC_{50}$ values ranged from 4.7 to 6.9 (see Supplementary Table 2). Simultaneously, compound purity and mass were analysed using LC–MS.

### Activity-based protein profiling guides hit selection

Subsequently, the qualified hits underwent testing in an orthogonal gel-based competitive activity-based protein profiling (ABPP) assay, a powerful technique for obtaining an initial selectivity profile of these compounds. The assay employed fluorophosphonate-tetramethylrhodamine (FP-TAMRA), a broad-spectrum activity-based probe designed to target a wide range of serine hydrolases, including endocannabinoid hydrolases like MAGL, fatty acid amide hydrolase (FAAH), and alpha/beta hydrolase domain-containing protein 6 (ABHD6)[30,31]. Using FP-TAMRA (100 nM, 10 min incubation) on mouse brain membrane proteome resulted in the identification of seven compounds showing >25% MAGL inhibition at 10 μM under these conditions (Fig. 1d, e and Supplementary Fig. 1). Interestingly, certain chemotypes, specifically benzoxazines and naphthyl amides, exhibited significant inhibition in the primary assay but proved ineffective in the ABPP assay. This observation could indicate potential species differences or a short residence time for the target. A correlation between inhibitor potency in the primary substrate-based assay and the orthogonal ABPP assay was, however, noted for the majority of

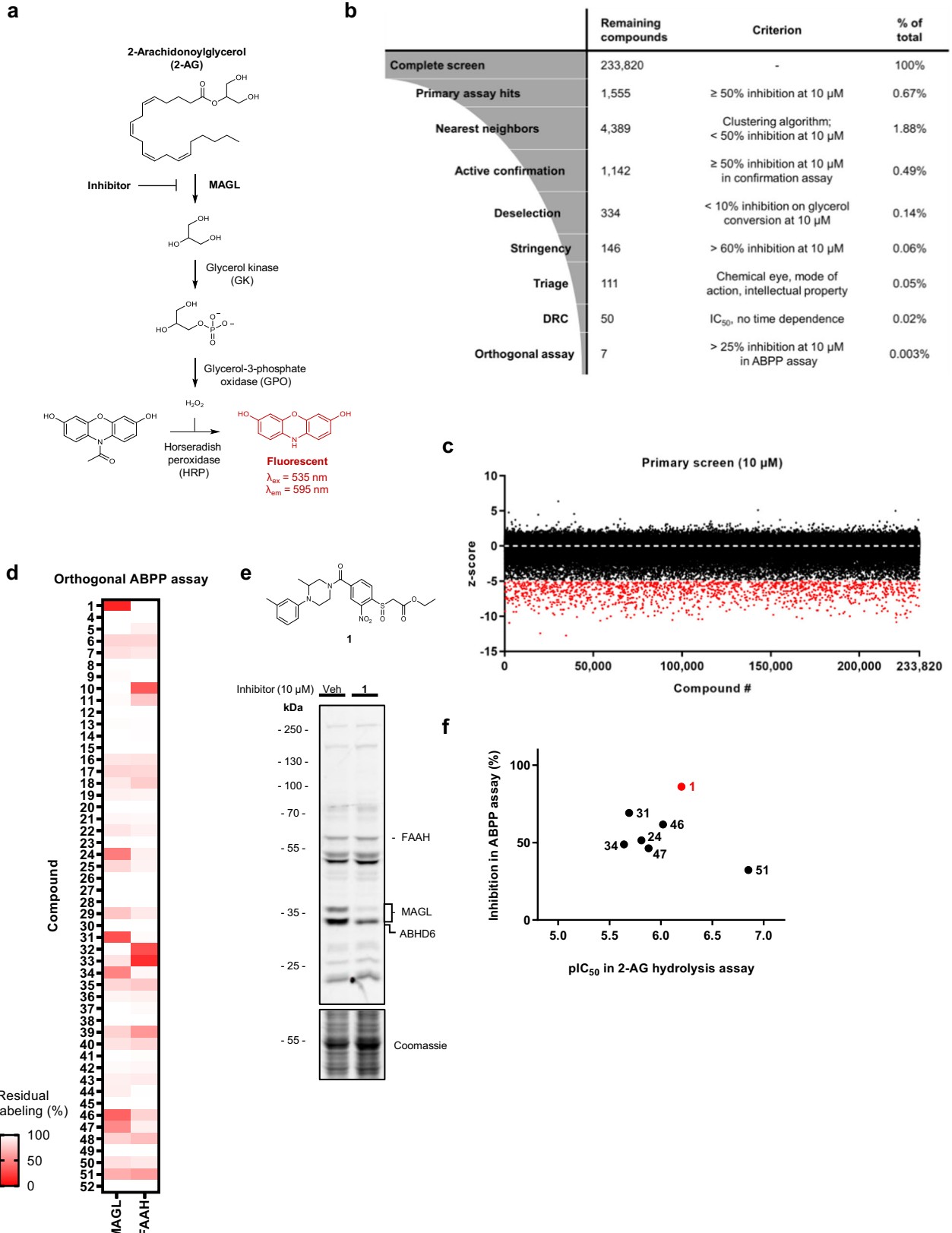

**Fig. 1 | Identification of reversible MAGL inhibitors in a high-throughput screen guided by activity-based protein profiling. a** MAGL activity assay, employing an enzyme cascade reaction to couple 2-AG hydrolysis to the generation of a fluorescent signal. **b** HTS triage process. Criteria and cut-offs at each step are indicated. **c** Results from the primary screen, expressed as z-score per individual compound. Compounds showing ≥50% inhibition (z-score ≤ −4.95) are shown in red. **d** Hit validation and FAAH selectivity assessment based on orthogonal ABPP assay. **e** Structure and competitive ABPP labelling profile of compound 1 on mouse brain proteome labelled by FP-TAMRA (n = 1). **f** Correlation between primary 2-AG hydrolysis activity assay and orthogonal competitive ABPP assay. Corresponding ABPP gels are shown in Supplementary Fig. 1. Complete HTS data are listed in Supplementary Tables 1 and 2.

compounds (Fig. 1f). The hits displayed, in general, good selectivity profiles on the mouse brain membrane proteome at 10 μM. Though several compounds demonstrated more than 25% inhibition of FAAH, which is responsible for the hydrolysis of the endocannabinoid anandamide (AEA, Fig. 1d). Upon direct comparison of MAGL versus FAAH labelling profiles, it was evident that compound **1** (see Fig. 1e) emerged as the most potent MAGL inhibitor and demonstrated selectivity over FAAH. This characteristic positions it as a promising candidate for further optimization.

## Hit optimization results in the discovery of MAGL inhibitor LEI-515

On resynthesis, chemical analysis and retesting of compound **1** (ethyl 2-((4-(3-methyl-4-(m-tolyl)piperazine-1-carbonyl)-2-nitrophenyl)sulfinyl)acetate), the identity and activity (pIC$_{50}$ = 6.4 ± 0.1) of the hit was confirmed (Fig. 2a). To improve the inhibitory activity of β-sulfinyl ester **1** a medicinal chemistry programme was initiated in which over 100 derivatives were synthesized and tested in the natural substrate and ABPP assays. A full report of the medicinal chemistry activities is described elsewhere[32]; however, in brief, we used a ligand-based drug design approach to improve the potency of compound **1**. In addition, our goal was to substitute the nitrophenyl group, which is a potential toxicophore. This led to the discovery of compound ±**2** as a potent MAGL inhibitor with single-digit nanomolar potency (pIC$_{50}$ = 8.5 ± 0.1)

(Fig. 2a and b). Compound ±**2** contains an ester functionality, which is a metabolic liability that limits the in vivo application of inhibitors. From our structure–activity relationship study, it became clear, however, that the ester was important for the inhibitory activity of the compound. We hypothesized that the carbonyl of the ester could act as an electrophile for the incoming catalytic serine of MAGL. We substituted, therefore, the ester with an α-CF$_2$ ketone as a bioisostere that may function as a warhead, which is resistant towards hydrolysis by esterases. This led to the discovery of LEI-515 (**3**) as a potent, metabolically stable MAGL inhibitor with a pIC$_{50}$ = 9.3 ± 0.1 (Fig. 2a and b). A detailed description of the synthesis of compound **1**, ± **2** and LEI-515 can be found in the supplementary information.

## Crystallization study reveals covalent binding pose of LEI-515

LEI-515 is thought to inhibit MAGL via forming a covalent reversible, enzyme-inhibitor hemiketal adduct (Fig. 2c). To investigate this hypothesis, we determined the crystal structure of human recombinant MAGL and in complex with racemic LEI-515. The structure of one isomer of LEI-515 ((1-((2-chloro-4-((2S,3S)-4-(3-chlorophenyl)-2,3-dimethylpiperazine-1-carbonyl)phenyl)sulfinyl)-3,3-difluoropentan-2-one) bound to MAGL was obtained and solved at 1.55 Å resolution (Supplementary Table 3, PDB-code: 8AQF). LEI-515 was covalently attached to the catalytic Ser122 residue through its electrophilic carbonyl group as a deprotonated hemiketal (Fig. 2d and e). The negative

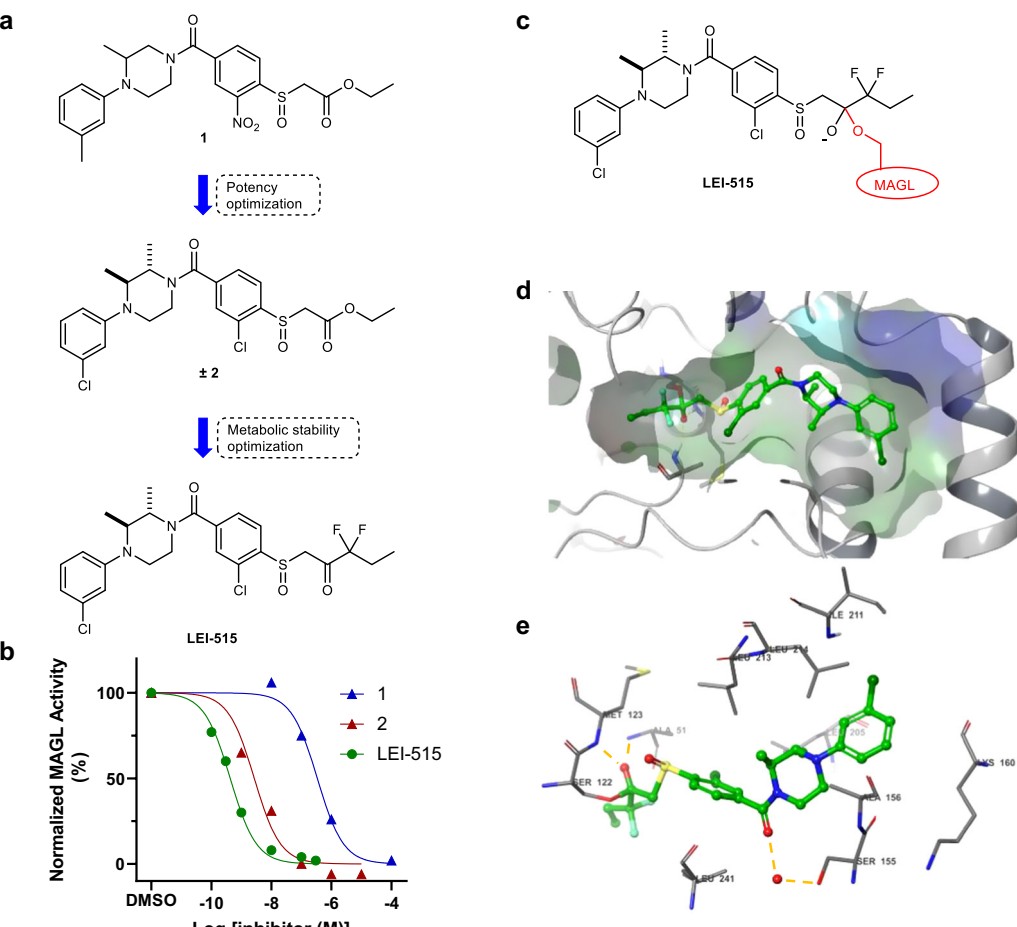

**Fig. 2 | Discovery of LEI-515 and its binding pose in human MAGL. a** Hit optimization of compound 1 led to the discovery of LEI-515. Substitution of the ester moiety for CF$_2$-ketone moiety significantly improved the metabolic stability. **b** Proposed transition state of LEI-515 bound to MAGL. The catalytic Ser122 of MAGL covalently interacts with the ketone group of LEI-515. **c** MAGL in vitro activity assay dose-response curves for Hit **1** (pIC$_{50}$ = 6.4 ± 0.1), **2** (pIC$_5$ = 8.5 ± 0.1) and LEI-515

(pIC$_{50}$ = 9.3 ± 0.1). Data represented as mean values (N = 2, n = 2). **d** and **e** The X-ray structure of (2S,3S)-isomer of LEI-515 bound to hMAGL. LEI-515 binds to MAGL through a covalently reversible mechanism and the deprotonated hemiketal forms two hydrogen bounds (yellow dotted line) with Ala51 and Met123. Source data are provided as a Source Data file.

charge was stabilized by two hydrogen bonds (yellow dotted lines) with Ala51 and Met123, respectively. The terminal ethyl group was inserted into the hydrophilic cytoplasmic access (CA) channel, while the rest of the compound occupied the large hydrophobic tunnel and the terminal chlorophenyl ring orientated to the lid domain. The chloro group on the other phenyl occupied a hydrophobic subpocket and the two methyl groups on the piperazine ring adopted a di-axial conformation, which enhanced the van der Waals interactions. The carbonyl of the amide group formed a water-mediated hydrogen bond with the side chain of Ser155.

## LEI-515 is a covalent reversible MAGL inhibitor

To study whether LEI-515 also inhibited endogenous MAGL, we incubated the compound or DMSO in brain homogenates and measured 2-AG levels by LC–MS/MS. It was found the LEI-515 increased 2-AG levels with similar potency in both human and mouse brain lysates (Fig. 3a). To confirm the covalent reversible mode-of-action of LEI-515, mouse brain membrane proteome was pre-incubated with LEI-515 or the irreversible MAGL inhibitor ABX-1431 at the $IC_{80}$ concentration and the remaining MAGL activity was visualized with an irreversible MAGL-specific probe LEI-463-Cy5[33] in a time-dependent manner (Fig. 3b). No recovery of MAGL activity was found for ABX-1431, whereas MAGL activity could be time-dependently regained after treatment with LEI-515. This indicated that LEI-515 is a reversible MAGL inhibitor. Next, the selectivity of LEI-515 was assessed with a gel-based ABPP method, using broad spectrum of serine hydrolases probes FP-TAMRA and MB064 (Fig. 3c). LEI-515 reduced MAGL activity in a dose-dependent manner with $IC_{50}$ of 25 nM in mouse brain proteome and displayed >500-fold selectivity over diacylglycerol lipase (DAGL-α) and α/β-hydrolase domain containing 6 and 12 (ABHD6 and 12), which are enzymes involved in 2-AG biosynthesis and degradation, respectively. Moreover, LEI-515 did not inhibit FAAH. Mass spectrometry (MS)-based chemical proteomics experiments emulated these findings (Fig. 3d). MAGL was fully inhibited in various tissues. Of note, hormone-sensitive lipase (HSL) was identified as main off-target in lung, liver, and brain (Fig. 3d; Supplementary fig. 2), which was confirmed by gel-based ABPP experiments using membrane fractions of HEK293T cells expressing recombinant human or mouse HSL (Supplementary Fig. 3). Next, LEI-515 was profiled for general off-target pharmacology in a binding/functional panel comprising a panel of 44 enzymes, transporters, receptors, and ion channel targets (CEREP) (Table S4). LEI-515 showed >100-fold selectivity over those ion channels, receptors, and enzymes, including the cannabinoid receptors ($CB_1R$ and $CB_2R$), hERG channel and cyclooxygenases.

Having established that LEI-515 is a covalent, reversible MAGL inhibitor, we then investigated whether LEI-515 inhibits endogenous MAGL in living cells. First, we assessed cellular target engagement using a gel-based ABPP assay with tailor-made activity-based probe LEI-463-Cy5 (Fig. 3e). LEI-515 prevented the labelling of MAGL by LEI-463-Cy5 with a $pEC_{50}$ of 6.78 ± 0.11. Next, HS578t cells, which express MAGL protein and activity as determined by western blot and ABPP (Supplementary Fig. 4), were used to study cellular target modulation using targeted lipidomics. The cells were incubated with LEI-515 and the cellular 2-AG, AA and anandamide levels were determined with liquid chromatography–mass spectrometry/mass spectrometry (LC–MS/MS). LEI-515 significantly increased 2-AG levels in a time and concentration-dependent manner (Fig. 3f and g), whereas AA and anandamide levels were decreased (Fig. 3h and i).

## LEI-515 is a peripherally restricted, orally bioavailable MAGL inhibitor

Before testing whether LEI-515 possessed in vivo efficacy, the absorption, distribution, metabolism, and excretion (ADME) profile of LEI-515 was determined. LEI-515 demonstrated acceptable physicochemical properties for oral bioavailability (MW = 531 Da, cLogP = 4.7 and tPSA = 58 Å²). LEI-515 showed high stability (100% remaining after

180 min) in both human and mouse plasma. Clearance was moderate in human microsomes (30.9 µL min⁻¹ mg⁻¹) and low in mouse microsomes (<3.4 µL min⁻¹ mg⁻¹). LEI-515 exhibited high human and mouse plasma protein binding (99.6% for both). LEI-515 showed negligible cell permeability in Caco-2 cells ($P_{app}$A-B < 0.01 × 10⁻⁶ cm s⁻¹ and $P_{app}$B-A < 0.005 × 10⁻⁶ cm s⁻¹). This might be due to transporter proteins that efflux LEI-515 out of the cells. Cellular permeability with LLC-PK1 cells overexpressing mouse mdr-1 (P-gp) could not be determined due to recovery below 50% but the in vitro-parallel artificial membrane permeability assay (PAMPA) indicated a moderate to high intrinsic permeability of 9.6 × 10⁻⁶ cm s⁻¹.

In vivo pharmacokinetic analysis with LEI-515 dosed at 10 mg kg⁻¹ intravenously (i.v.) and oral (p.o.) in male C57BL/6J mice using DMSO/Kolliphore/5% mannitol in water (1/1/8, v/v) as vehicle revealed a moderate clearance (Cl = 35 mL min⁻¹ kg⁻¹) and volume of distribution ($V_{ss}$ = 2.1 L kg⁻¹), resulting in a half-life of 4.5 h (Supplementary Table 5). LEI-515 showed excellent oral bioavailability ($F_{po}$ = 81%) and quick absorption ($T_{max}$ = 0.5 h) (Fig. 4a). The high oral bioavailability was unexpected based on the Caco-2 data but could be explained by rapid absorption in the stomach. Of note, the brain-to-plasma ratio was 0.01 at maximum serum concentration ($C_{max}$), which indicates that LEI-515 is a peripherally restricted MAGL inhibitor.

In view of the encouraging pharmacokinetic properties and low brain exposure, we administered LEI-515 or vehicle to C57BL/6J mice (30 and 100 mg kg⁻¹, p.o., single dose) and determined target modulation in peripheral and brain tissue after 1 h. To this end, we analysed 2-AG, arachidonic acid (AA) and anandamide levels in mouse intestine, lung, liver and brain by LC-MS/MS. LEI-515 induced a concentration-dependent increase in 2-AG in the intestines and lungs, but not in the brain. In the liver, a significant correlation was found between increased 2-AG levels and LEI-515 concentration (Fig. 4b, Supplementary Fig. 5). Anandamide and AA levels were not affected (Fig. 4c and d). Collectively, these data indicate that LEI-515 is an in vivo active MAGL inhibitor that is restricted to the periphery and can be used to study the physiological role of MAGL outside the central nervous system.

## LEI-515 displays efficacy in acute liver injury model

Previously, the CNS penetrant, covalent irreversible MAGL inhibitor JZL184 was able to attenuate acute liver injury induced by $CCl_4$[9]. Here, we wanted to assess whether our reversible MAGL inhibitor LEI-515 was able to replicate these findings. 24 h following administration, $CCl_4$ induced massive necrosis in the liver evidenced by marked elevations of the liver transaminases (AST and ALT) and histopathological injury. LEI-515 (30 mg/kg, i.p.) significantly reduced $CCl_4$-induced liver transaminase elevation and histopathological injury (Fig. 5a–f). LEI-515 had no effect in control animals (Supplementary Fig. 6a–c). $CCl_4$-induced massive CD45 positive leucocyte infiltration (brown staining) around the necrotic area in the liver, which was significantly attenuated by the LEI-515 treatment (Fig. 5d and e). LEI-515 had no effect in control animals on liver inflammation (Supplementary Fig. 6d and e). $CCl_4$-induced marked increase in the hepatic accumulation of 4-HNE (brown staining), a marker of lipid peroxidation/oxidative stress (Fig. 5f and g). LEI-515 significantly reduced $CCl_4$-induced hepatic oxidative stress, without having any effect in the liver of control mice (Fig. 5f, g and Supplementary Fig. 6f, g). Thus, these findings demonstrated that LEI-515 attenuated liver necrosis, inflammation, and oxidative stress in an acute liver injury model.

## LEI-515 displays cannabinoid $CB_2$ receptor-dependent antinociceptive efficacy in a chemotherapy-induced neuropathic pain model

2-AG is implicated in the regulation of pain sensation[34]. Consequently, centrally acting MAGL inhibitors exhibit antinociceptive properties in various preclinical pain models[35]. We examined the impact of LEI-515 on behavioural sensitization to mechanical stimulation that was induced by the taxane chemotherapeutic agent paclitaxel in mice of

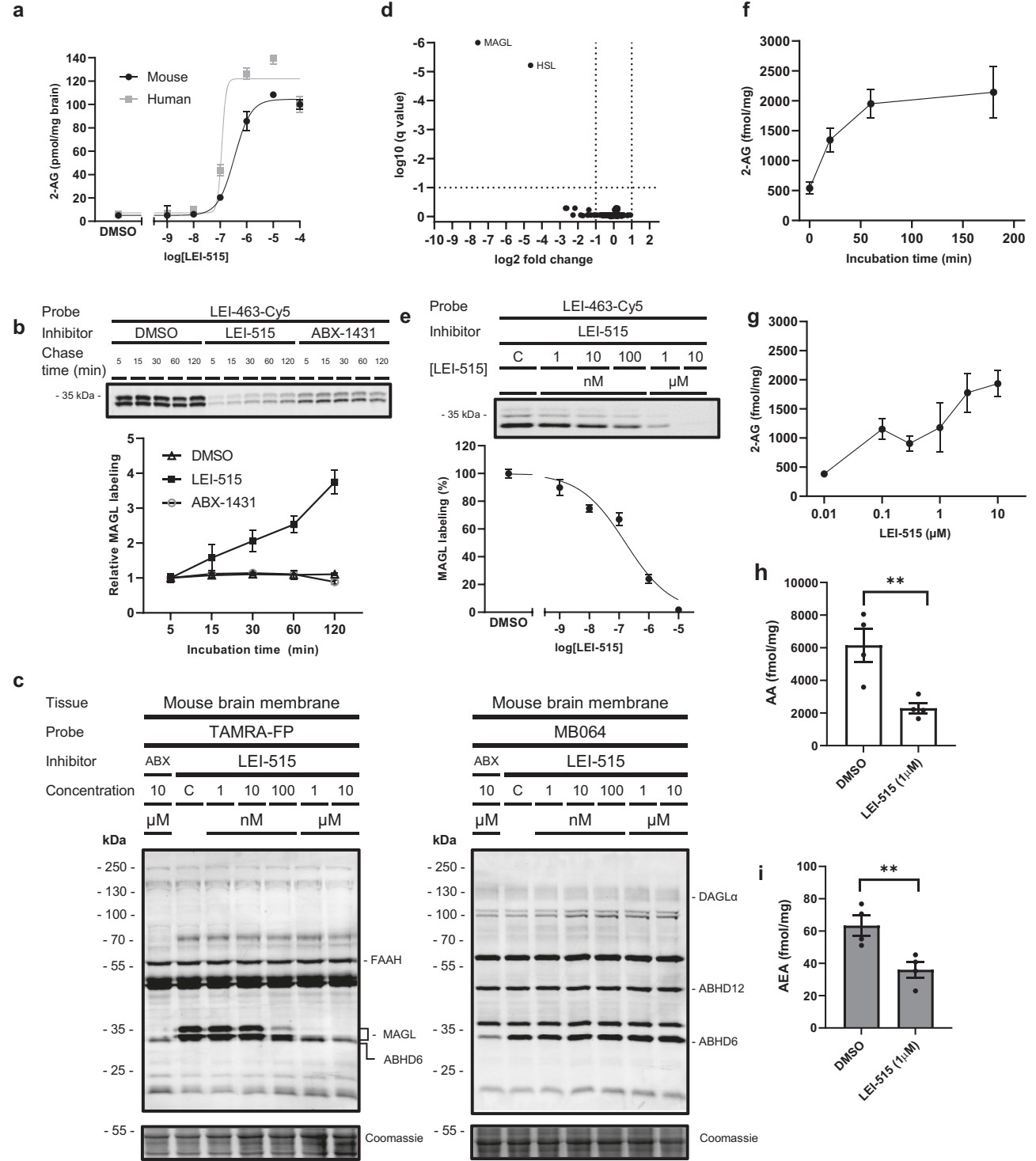

**Fig. 3 | LEI-515 is a reversible MAGL inhibitor and active in cells. a** LEI-515 increases 2-AG concentration in mouse (pIC$_{50}$ = 6.6) and human brain lysates (pIC$_{50}$ = 7.0). **b** Time-dependent recovery of MAGL activity in mouse brain proteome of LEI-515 and ABX-1431 as determined by competitive ABPP using a MAGL selective probe LEI-463-Cy5 (1 μM). Data represented as the fold change in MAGL labelling of the two isoforms as compared to the 5 min chase time. $n$ = 3 biologically independent samples. **c** Dose-dependent gel-based ABPP profile of LEI-515 in mouse brain membrane proteome using broad spectrum serine hydrolase probes FP-TAMRA (100 nM) and MB064 (250 nM). C = DMSO control, ABX = ABX-1431 ($n$ = 2). **d** Chemical proteomics-based selectivity profiles of LEI-515 (1 μM, 30 min, 37 °C) on mouse lung proteome using a probe cocktail of MB108 and FP-Biotin (2.5 μM and 5 μM, respectively, 30 min, 37 °C). Data represented as a volcano plot with cut-off values:

unique peptides ≥2; −1 ≤ log$_2$(fold change) ≥ 1; $p$ < 0.05 using multiple unpaired $t$-test with FDR 5% ($n$ = 4). **e** Dose-dependent cellular target engagement of LEI-515 (pEC$_{50}$ 6.78 ± 0.11) in U-87 MG live cells using LEI-463-Cy5 (10 nM). $n$ = 3 biologically independent samples. **f** In situ treatment of HS578t cells with LEI-515 (1 μM) time-dependently increased cellular 2-AG levels. $n$ = 5 biologically independent samples. **g** LEI-515 (1 h) dose-dependently increased cellular 2-AG levels. $n$ = 4 biologically independent samples. **h** and **i** In situ treatment of HS578t cells with LEI-515 (1 μM, 1 h) decreased cellular arachidonic acid (AA, $p$ = 0.0011) (**h**) and anandamide ($p$ = 0.0029) (AEA, **i**) levels. $n$ = 4 biologically independent samples. Statistical analysis: one-way ANOVA with Dunnett's post-hoc test (***$p$ < 0.001, **$p$ < 0.01, *$p$ < 0.05 vs. vehicle). All data are represented as mean values ± SEM. Source data are provided as a Source Data file.

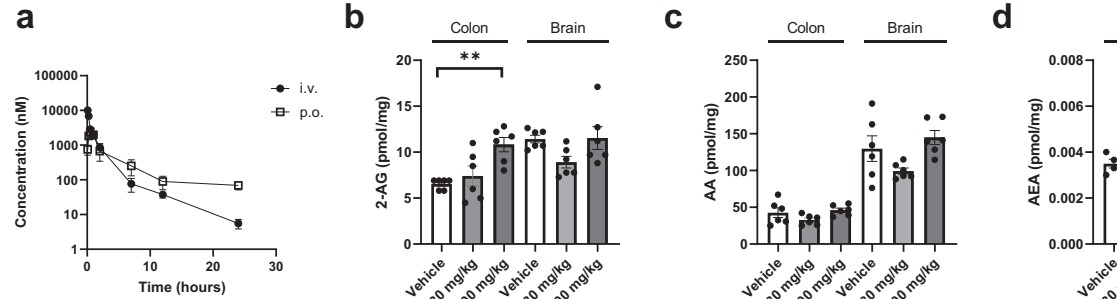

**Fig. 4 | LEI-515 is an orally active, peripherally restricted MAGL inhibitor. a** In vivo pharmacokinetics of LEI-515 in plasma of C57BL/6J mice via intravenous (i.v., 10 mg/kg) or oral (p.o., 10 mg/kg) administration ($n = 3$). **b–d** Lipid levels of **b** 2-AG (colon 100 mg/kg, $p = 0.0003$), **c** AA and **d** anandamide (AEA) in C57BL/6J mice brain and colon after oral administration (30 or 100 mg/kg) of LEI-515. Statistical analysis: Unpaired two-tailed $t$-test (***$p < 0.001$, **$p < 0.01$, *$p < 0.05$ vs. vehicle; $n = 6$). Data are presented as mean ± SEM. Source data are provided as a Source Data file.

both sexes (Fig. 6a–f). LEI-515 dose-dependently suppressed established paclitaxel-induced mechanical hypersensitivity in both male and female mice following oral administration; the $ED_{50}$ (95% confidence intervals) was 1.01 (0.31–3.30) and 1.41 mg/kg (0.30–6.63) mg/kg p.o. in male and female mice, respectively and the 95% confidence intervals overlapped between sexes (Fig. 6a). Efficacy of LEI-515 peaked at 2 h and was still present 24 h following acute oral gavage (Fig. 6b and Supplementary Fig. 7a). Antinociceptive efficacy of LEI-515 (p.o.) was fully sustained following repeated dosing over 10 consecutive days (Fig. 6c and Supplementary Fig. 7c); residual antinociceptive efficacy was still present 4 days following termination of repeated p.o. dosing (Fig. 6c). LEI-515 also produced dose-dependent suppressions of paclitaxel-induced mechanical hypersensitivity following i.p. administration; the $ED_{50}$ (95% confidence intervals) was 1.09 mg/kg (0.26–4.61) mg/kg and 0.20 (0.02–2.52) mg/kg i.p. in male and female mice, respectively, and the 95% confidence intervals overlapped between sexes (Fig. 6d). Efficacy of acute LEI-515 was maximal at 2 h following i.p. injection in both sexes (Fig. 6e and Supplementary Fig. 7b). Antinociceptive efficacy of LEI-515 was also observed 24 h after acute (i.p.) dosing and was fully sustained following 10 days of once daily i.p. dosing (Fig. 6f and Supplementary Fig. 7d); residual antinociceptive efficacy was observed 4 days following termination of chronic dosing (Fig. 6f). By contrast, tolerance developed to the brain penetrant MAGL inhibitor JZL184[36] as well as orthosteric $CB_1$ agonists $\Delta^9$-tetrahydrocannabinol[37], CP55,940[38] and WIN55,212-2[39] in our laboratory in the same model under identical conditions. Pharmacological specificity of LEI-515 antinociceptive efficacy was assessed in paclitaxel-treated mice using cannabinoid $CB_1$ (AM251 and AM6545) and $CB_2$ (AM630 and SR144528) receptor antagonists that differ in their ability to penetrate the CNS. Efficacy of LEI-515 in suppressing paclitaxel-induced neuropathic nociception was fully blocked by either CNS penetrant (AM630) or peripherally restricted (SR144528) $CB_2$ receptor antagonists[40] (Fig. 7a) but not by CNS penetrant (AM251) or peripherally restricted (AM6545) $CB_1$ receptor antagonists (Fig. 7b). These observations suggest that the therapeutic profile of LEI-515 is mediated by a peripheral $CB_2$ receptor mechanism. Furthermore, LEI-515 did not alter mechanical paw withdrawal thresholds in control mice that received the cremophor-based vehicle in lieu of paclitaxel (Fig. 7c). Thus, LEI-515 reversed chemotherapy-induced neuropathic nociception without altering basal nociceptive thresholds.

To further examine whether LEI-515 or its metabolites would induce any cannabinoid $CB_1$ receptor-dependent CNS-side effects, we tested the compound in a triad of behavioural assays that are highly sensitive to $CB_1$ receptor activation (ring immobility, rectal temperature, and hot water tail-flick assays). The global MAGL inhibitor JZL184 and the non-selective $CB_1$ receptor agonist WIN55,212-2 were used as comparators. As shown in Fig. 8a–c, WIN55,212-2 elicited catalepsy,

hypothermia, and tail-flick antinociception in a time-dependent manner, whereas JZL184 trended to increase ($p = 0.096$) time spent immobile in the ring test 2 h after i.p. administration. In contrast, LEI-515 did not produce any detectable catalepsy, hypothermia or antinociception in the hot water tail-flick assay, a test of spinally mediated antinociception, at any time point (Fig. 8a–c). These findings are consistent with the peripherally restricted nature of LEI-515. Next, we asked whether a challenge with the $CB_1$ antagonist rimonabant would evoke signs of physical dependence in mice treated chronically for 20 consecutive days with LEI-515, JZL184 and WIN55,212-2 (Fig. 8d–g). As expected, rimonabant challenge evoked paw tremors (Fig. 8d) and head twitches (Fig. 8e) in mice treated chronically with JZL184 and WIN55,212-2, effects which were absent in mice treated chronically with LEI-515 or vehicle. Rimonabant challenge also evoked prominent withdrawal jumps in mice treated chronically with JZL184; these effects were also observed in a small subset of mice treated with WIN55,212-2 but not in mice treated with LEI-515 (Fig. 8f). Rimonabant challenge also evoked scratching behaviours in mice treated chronically with vehicle or LEI-515; this scratching behaviour was attenuated in mice treated chronically with either JZL184 or WIN55,212-2 (Fig. 8g). By contrast, rimonabant-induced scratching did not differ between groups treated chronically with either LEI-515 or vehicle (Fig. 8g). Thus, on dependent measures identifying CNS side-effects or physical dependence, effects of chronic dosing with LEI-515 could be differentiated from treatment with the orthosteric cannabinoid agonist WIN55,212-2 or the global MAGL inhibitor JZL184, and was also indistinguishable from chronic dosing with the vehicle. To summarize, our studies suggest that LEI-515, a peripherally restricted MAGL inhibitor, shows promise as a therapeutic strategy for attenuating tissue injury and chemotherapy-induced neuropathic pain without producing physical dependence or unwanted pharmacological effects associated with direct $CB_1$ receptor activation.

## Discussion

Cannabinoids exert antinociceptive properties by activating cannabinoid $CB_1$ and/or $CB_2$ receptors. Several cannabis-based drugs, such as Cesamet®, Marinol® and Sativex®, have been clinically approved to reduce neuropathic pain in Canada and some European countries. Yet, activation of the cannabinoid $CB_1$ receptor by these drugs is associated with unwanted psychotropic side effects that limit their clinical utility. Indirect activation of the cannabinoid receptor, by raising endocannabinoid levels through blocking their metabolism, has been proposed as a suitable alternative therapeutic option without inducing full-blown cannabis-like psychotropic effects. Chemotherapy-induced peripheral neuropathy is a dose-limiting adverse effect of anti-tumour agents (e.g. cis-platin, paclitaxel) used in the treatment of breast, lung and colorectal colon cancer, and represents a major reason for

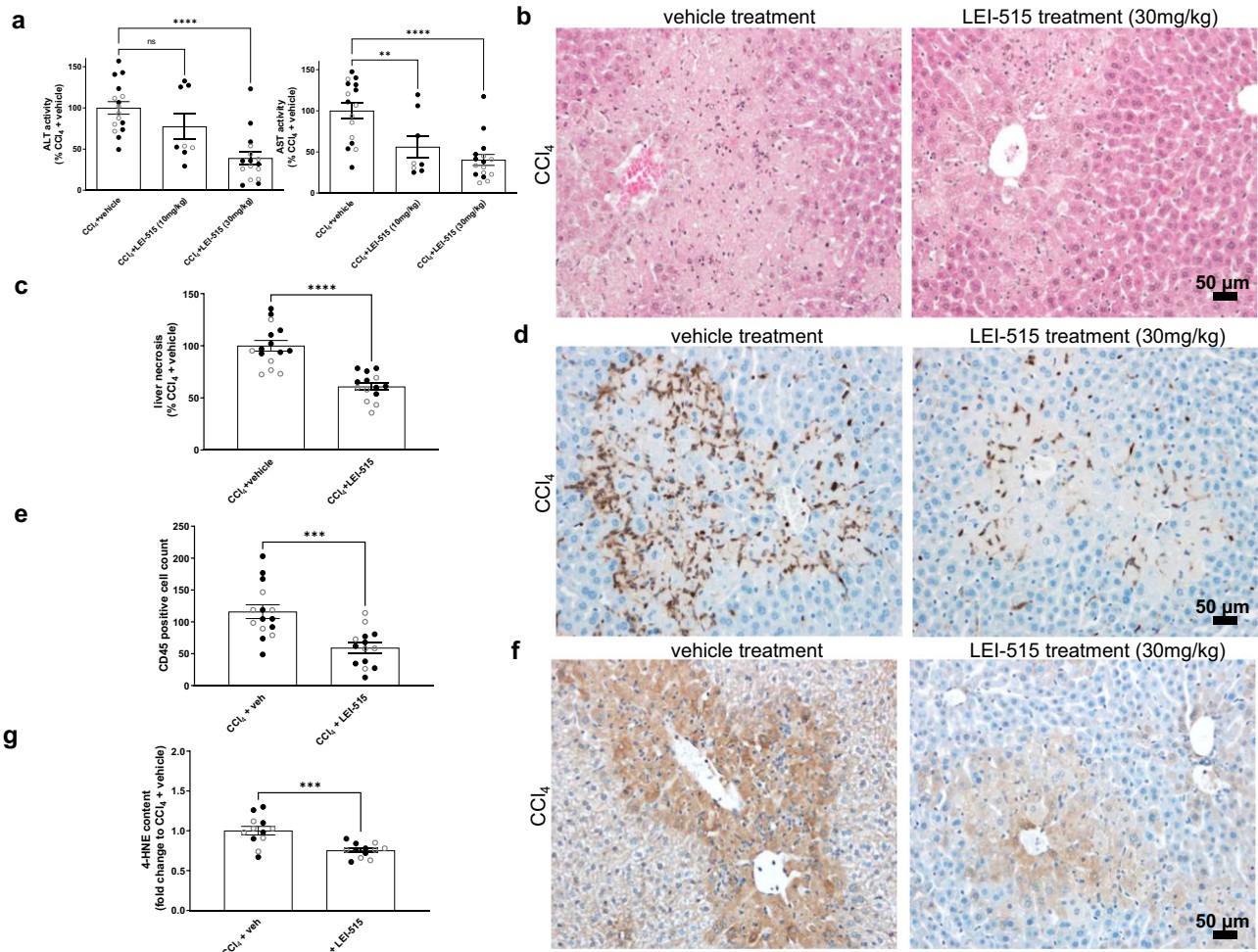

**Fig. 5 | LEI-515 reduces acute liver injury. a** Effect of acute $CCl_4$ treatment with vehicle ($n = 16$) or LEI-515 (10 and 30 mg/kg i.p., $n = 8$ and $n = 16$/group, respectively) on liver transaminases [alanine aminotransferase (ALT, $p < 0.0001$) and aspartate aminotransferase (AST, $p = 0.0084$ and $<0.0001$, respectively)], **b** and **c** histopathological liver necrosis and its quantification ($n = 15$ in $CCl_4$+veh and $n = 14$ in $CCl_4$+LEI 30 mg/kg groups, respectively; $p < 0.0001$), **d** and **e** CD45+ leucocyte infiltration and its quantification ($n = 15$ in $CCl_4$+veh and $n = 14$ in $CCl_4$+LEI 30 mg/kg groups, respectively; p = 0.0002), **f** and **g** lipid peroxidation (4-HNE) 24 h following $CCl_4$ administration and its quantification ($n = 12$/group, p = 0.0004). All data are presented as mean ± SEM. Open symbols represent females, whereas closed symbols show male subjects. Statistical analysis: One-way ANOVA and Dunnett post-hoc test (**a**), unpaired two-tailed $t$-test (**c**, **e**, **g**) (****$p < 0.0001$, ***$p < 0.001$, **$p < 0.01$ vs. vehicle; $n = 8$–16/group). The drug had a comparable effect in both male and female mice. Source data are provided as a Source Data file.

discontinuation of chemotherapy treatment[41]. Iatrogenic side effects include sensory abnormalities and pain that can persist long after treatment cessation[42,43]. Currently, available analgesic drugs show limited efficacy, hampering the management of chemotherapy-induced neuropathy[44]. There is a significant unmet medical need to identify safe and effective analgesics for treating chemotherapy-induced neuropathic pain. Indeed, centrally acting MAGL inhibitors JZL184 and MJN110 attenuated mechanical and cold hypersensitivity in a chemotherapy-induced neuropathy mouse model via cannabinoid $CB_1$ and $CB_2$ receptor-dependent mechanisms[45,46]. However, chronic dosing with these centrally active MAGL inhibitors resulted in rapid tolerance and was associated with cannabinoid $CB_1$ receptor-dependent withdrawal symptoms[15]. These findings clearly indicate the importance of assessing the biological consequences of increasing 2-AG levels only outside the CNS. However, this represents an important challenge, because of a lack of peripherally restricted MAGL inhibitors.

Here, we have addressed this unmet clinical need by developing LEI-515 as a reversible, peripherally restricted inhibitor of MAGL. A high throughput screening campaign followed by a medicinal chemistry programme in which potency, selectivity and metabolic stability were optimized led to the discovery of LEI-515. Key to the development of LEI-515 was the use of ABPP to select the best confirmed hit and to guide its optimization. LEI-515 is a potent and reversible MAGL inhibitor as demonstrated by our biochemical, ABPP and cellular assays. It makes use of an activated ketone as a warhead that covalently interacted with the catalytically active Ser-122 as witnessed in the co-crystal structure with human MAGL. LEI-515 increased 2-AG concentrations in mouse and human brain lysates and in cells. AA and anandamide levels were decreased in cells as a consequence of MAGL inhibition. Since LEI-515 does not inhibit NAPE-PLD and AA is required for the production of anandamide, we interpret the reduced anandamide levels as a consequence of lowered AA levels (i.e. an indirect substrate–product relationship). In our studies, oral administration of LEI-515 dose-dependently increased 2-AG levels in the periphery without altering anandamide levels. Importantly, LEI-515 levels in the brain were 100-fold lower than the $C_{max}$ in plasma. Consequently, 2-AG and AA levels were not affected in the brain. The peripheral restriction of LEI-515 is most likely the result of a combination of high lipophilicity, high plasma protein binding and PgP-transporter efflux. LEI-515 thus provided us the opportunity to investigate the biological role of MAGL in tissue injury and pathological nociception in the periphery.

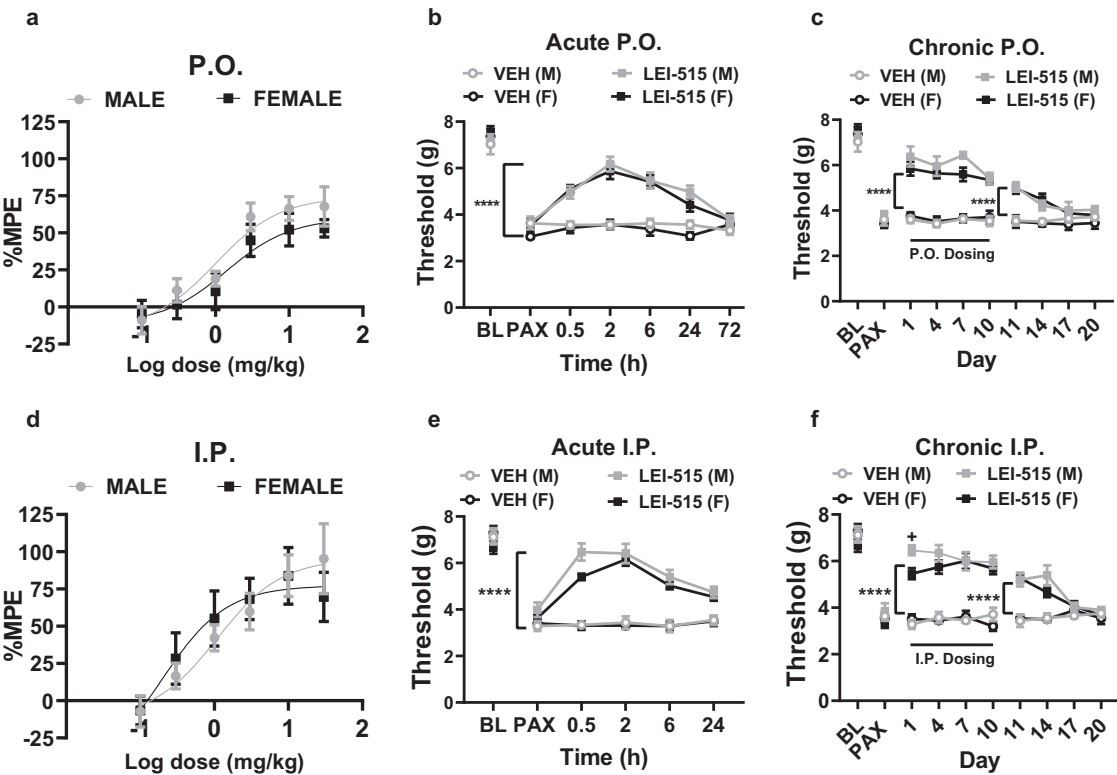

**Fig. 6 | Acute and chronic dosing with peripherally restricted MAGL inhibitor LEI-515, administered p.o. or i.p., suppresses paclitaxel-induced mechanical hypersensitivity in mice of both sexes. a** LEI-515 (0.1, 0.3, 1, 3, 10 and 30 mg/kg p.o.) increases percent maximal antinociceptive effect (%MPE) relative to pre-paclitaxel baseline in male ($n = 7$) and female ($n = 8$) mice. $ED_{50}$ (95% confidence intervals): 1.01 (0.31–3.30) mg/kg and 1.41 (0.30–6.63) mg/kg p.o. in male and female mice, respectively. **b** Acute LEI-515 (10 mg/kg p.o.) suppressed paclitaxel-induced mechanical hypersensitivity in male and female mice with efficacy lasting over 24 h. ****$p < 0.0001$ LEI-515 vs. vehicle control groups. **c** Chronic dosing with LEI-515 (10 mg/kg/day p.o. for 10 days) suppressed paclitaxel-induced hypersensitivity without loss of efficacy in both sexes. ****$p < 0.0001$ LEI-515 vs. vehicle groups, male LEI-515. **d** LEI-515 (0.1, 0.3, 1, 3, 10 and 30 mg/kg i.p.) increases %MPE relative to pre-paclitaxel baseline in male and female mice. $ED_{50}$ (95% confidence intervals): 1.09 (0.26–4.61) and 0.20 (0.02–2.52) mg/kg i.p. in male and female mice, respectively. $n = 8$ per group. **e** Acute LEI-515 (10 mg/kg i.p.) suppressed paclitaxel-induced

mechanical hypersensitivity in male and female mice with efficacy lasting over 24 h. $n = 7$–8 per group. **f** Chronic dosing with LEI-515 (10 mg/kg/day i.p. for 10 days) suppresses paclitaxel-induced mechanical hypersensitivity without loss of efficacy in both sexes. ****$p < 0.0001$ LEI-515 vs. vehicle control (Three-way ANOVA, Tukey post hoc test). LEI-515 trended to increase paw withdrawal thresholds to a greater extent in paclitaxel-treated males than females ($p = 0.0569$; Tukey post hoc test). No main effect of sex ($p = 0.206$) or other interactions (Sex × Time: $p = 0.2165$: Time × Treatment: $p = 0.8939$; Sex × Treatment: $p = 0.3299$) were detected. Thresholds were higher in LEI-515-treated males compared to female mice on day 1 of chronic dosing only (+$p = 0.0287$, Sidak's post hoc test) (see Supplementary Fig. 7d). Efficacy was preserved at least 4 days following termination of repeated i.p. dosing; ****$p < 0.0001$ LEI-515 vs. vehicle control. **b**, **c**, **e**, **f** $n = 7$–8 per group. All data are represented as mean values ± SEM. Statistical analysis: See Supplementary Tables S8 and S9 for Three-way and Two-way ANOVA results. Source data are provided as a Source Data file.

Previously, MAGL inhibition and/or genetic deletion has been shown to attenuate acute liver injury induced by ischaemia-reperfusion, GalN/LPS and CCl$_4$[9]. Irreversible MAGL-inhibition and/or genetic deletion of MAGL in mice attenuated hydrolysis of 2-AG to arachidonic acid and consequently pro-inflammatory prostaglandins in hepatocytes and inflammatory cells independently from CB$_2$ receptor activation. In parallel, it also attenuated endothelial activation, pro-inflammatory response in Kupffer cells and chemotaxis of infiltrating immune cells via CB$_2$ receptor activation[9]. Consistently with these results, we found that the reversible MAGL inhibitor LEI-515 markedly attenuated the CCl$_4$-induced acute liver injury, inflammation and oxidative stress, suggesting its potential utility in liver disease.

Having established that LEI-515 mimics the effects of irreversible MAGL inhibitors in a liver injury model, we investigated whether peripheral MAGL inhibition would suppress behavioural sensitization in a chemotherapy-induced neuropathic pain model without inducing CNS-mediated adverse effects. LEI-515 dose-dependently suppressed mechanical hypersensitivity in paclitaxel-treated mice of both sexes after either oral or i.p. administration. The effect of acute dosing was long-lasting (>24 h) and was maintained upon chronic dosing. By contrast, tolerance developed to the global MAGL inhibitor JZL184[36] as

well as orthosteric cannabinoid agonists (i.e. THC, WIN55,212-2, CP55,940)[37–39] in this model and also developed to the peripherally restricted CB$_1$ agonist CB-13[47]. Anti-allodynic efficacy of LEI-515 was blocked by CB$_2$ receptor antagonists that differ in their ability to penetrate the CNS, but not by global or peripherally restricted CB$_1$ receptor antagonists. It is unlikely that the doses of CB$_1$ antagonists employed were ineffective because these antagonist doses blocked the effects of global (URB597) and peripherally restricted (URB937) FAAH inhibitors in the same model in our previous work[39]. Our data indicate the involvement of a peripheral CB$_2$ receptor mechanism in the anti-allodynic effects of LEI-515, possibly in the dorsal root ganglia or skin. Using a CB$_2$-EGFP reporter mouse, we previously showed that CB$_2$ receptors are localized to peripheral sites (keratinocytes, Langerhans cells, dendritic cells, Merkel cell endings, spleen) and documented that the present paclitaxel dosing protocol produced dynamic regulation of CB$_2$ reporter in keratinocytes and Langerhans cells in the epidermis under conditions in which CB$_2$ was not detected using immunohistochemistry in CNS (using anti-GFP immunofluorescence as a surrogate marker for CB$_2$)[40]. Of note, CB$_2$ agonists that differ in their ability to penetrate the CNS (i.e. LY2828360 and AM1710 which exhibit high and low CNS penetration, respectively) produce CB$_2$-mediated

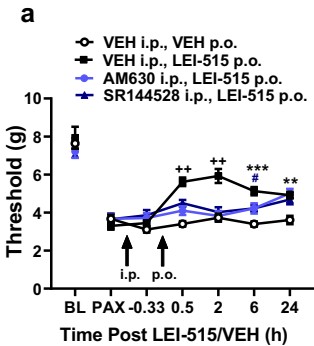
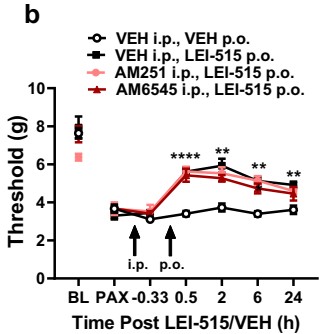
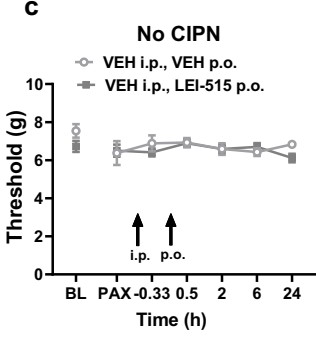

**Fig. 7 | The peripherally restricted MAGL inhibitor LEI-515 suppresses mechanical hypersensitivity in male mice through a peripheral CB₂ mechanism. a** LEI-515 (p.o.) suppressed paclitaxel-induced mechanical hypersensitivity in a manner that was blocked equivalently by pretreatment with CNS penetrant (AM630; 5 mg/kg i.p.) and peripherally restricted (SR144528; 2.1 mg/kg i.p.) CB₂ antagonists across the observation interval ($p < 0.0001$ for all comparisons). Treatment effects were also time dependent [++]$p < 0.01$, LEI-515 vs. all other groups; [***]$p < 0.001$, [**]$p < 0.01$ vs. vehicle; [#]$p<0.05$ vs. SR144528, LEI-515. $n = 6$ male mice per group. **b** LEI-515 (p.o.)-induced suppression of paclitaxel-induced mechanical hypersensitivity was not blocked ($p = 0.98$ vs. AM251, LEI-515; $p = 0.2387$ vs. AM6545, LEI-515) by pretreatment with either the CNS penetrant (AM251; 5 mg/kg i.p.) or peripherally restricted (AM6545; 10 mg/kg i.p.) CB₁ antagonists throughout the observation interval. Antinociceptive effects of LEI-515 were also time-dependent [****]$p < 0.0001$, [***]$p < 0.001$, [**]$p < 0.01$ LEI-515 vs. vehicle. $n = 6$ male mice per group. **c** Neither LEI-515 (10 mg/kg p.o.) nor vehicle (i.p.) pretreatment altered mechanical paw withdrawal thresholds in mice receiving cremophor-based vehicles in lieu of paclitaxel ($p = 0.4948$). Two-way ANOVA with Sidak's multiple comparison test. $n = 6$ male mice per group. All data are represented as mean values ± SEM. Statistical analysis: Two-way ANOVA, with Dunnett multiple comparison tests comparing LEI-515 to all other groups. Source data and statistical analyses are provided as a Source Data file.

anti-allodynic efficacy in this model[40]; these effects are preserved in CB₁ knockout (KO) mice, absent in CB₂ KO mice, and blocked by both brain penetrant (AM630) and peripherally restricted (SR144528). These observations are in line with LEI-515 suppressing chemotherapy-induced neuropathic nociception through a peripheral CB₂ analgesic mechanism. Previously, peripheral CB₁ receptors were shown to mediate antinociceptive actions of direct cannabinoid receptor agonists[48]. The prominent role of CB₂ as opposed to CB₁ in the antinociceptive effects of LEI-515 may suggest that 2-AG is produced on demand at a site where CB₂ receptors are preferentially localized[40]. In contrast, the antinociceptive action of centrally acting MAGL inhibitors, such as JZL184 and MJN110, was partly dependent on a CB₁ receptor mechanism and induced rapid tolerance. LEI-515 also lacked cardinal signs of CB₁ activation. As expected, the direct CB₁ receptor agonist (WIN55,212-2) demonstrated cannabimimetic activity in three assays (ring immobility, rectal temperature, and hot water tail-flick antinociception) whereas JZL184 trended to increase immobility time in the ring test of catalepsy. Notably, LEI-515 did not show any activity in any of the three assays. Finally, rimonabant challenge evoked signs of physical dependence (paw tremors, head shakes, withdrawal jumps) in mice treated chronically with JZL184 and/or WIN55,212-2, but not in mice treated chronically with LEI-515 or vehicle. To our knowledge, the prominent withdrawal jump phenotype observed in mice treated chronically with JZL184 has not been previously reported. As expected, rimonabant challenge produced scratching behaviour, consistent with pruritis, effects shown previously to be absent in CB₁ KO mice[38,49], and this behavioural phenotype was reduced by JZL184 and WIN55,212-2, but not by LEI-515 in our studies. Thus, peripheral inhibition of MAGL activity is sufficient to suppress aberrant behavioural sensitization in chemotherapy-induced neuropathy via a CB₂ receptor mechanism without inducing unwanted CNS-mediated cannabimimetic effects, tolerance or physical dependence.

LEI-515 displays high selectivity over many receptors, ion channels and enzymes, including cannabinoid CB₁ and CB₂ receptors, hERG, NAPE-PLD, COX1, COX2, FAAH, ABHD6, ABHD12, DAGLα and DAGLβ, as assessed by activity-based protein profiling and CEREP screening. However, we did identify hormone-sensitive lipase (HSL) as a main off-target. HSL may act as a DAG-lipase in intestines, adipose tissue, and testis, which may lead to the production of monoacylglycerols, including 2-AG, under certain conditions[50]. When interpreting the results of LEI-515, its HSL inhibitory activity should be considered. In both our CCl₄ and paclitaxel models, the effects of LEI-515 were emulated by other MAGL inhibitors and genetic KO models; thus, HSL plays a limited role, if any, in these pathophysiological conditions. In fact, LEI-515 may provide an opportunity to decouple the 'on demand' 2-AG signalling pool produced via DAGLα/β from basal 2-AG levels provided by HSL (i.e. the metabolic pool). If 2-AG production is mainly dependent on HSL, then it is expected that LEI-515 would be less effective in raising 2-AG levels. This may provide a potential explanation for why the effective dose of LEI-515 in the chemotherapy-induced pain model was 10-fold lower than the dose required to increase 2-AG levels in the intestine.

To conclude, we have discovered a peripherally restricted MAGL inhibitor that attenuates acute liver injury, inflammation, oxidative stress and chemotherapy-induced neuropathic nociception without inducing unwanted cardinal signs of CB₁ activation, tolerance, or physical dependence. Our studies could provide a rationale and framework for the future development of a novel class of non-steroidal, anti-inflammatory analgesics with reduced side effects.

## Methods
### Cloning, overexpression and membrane preparation
Full-length cDNA encoding human MAGL (GenBank ID: BC006230.2; obtained from Source Bioscience) or mouse or human LIPE (referred to as 'HSL' elsewhere in manuscript; GenBank IDs: BC021642 and BC070041; obtained from BioCat GmbH) was amplified by PCR and cloned into expression vector pcDNA3.1 in frame with a C-terminal FLAG-tag. All plasmids were isolated from transformed XL10-Gold competent cells (prepared using *E. coli* transformation buffer set; Zymo Research) using plasmid isolation kits following the supplier's protocol (Qiagen). Constructs were verified by Sanger sequencing (Macrogen).

Human embryonic kidney (HEK293T) cells were sourced from ATCC (catalogue number: CRL-3216) and subjected to routine screening for mycoplasma contamination. Cultures were discarded after 2–3 months of utilization. The cells were cultured at 37 °C under 7% CO₂ in high-glucose DMEM comprising phenol red, stable glutamine, 10% (v/v) high iron newborn calf serum (Seradigm), and penicillin and streptomycin (200 µg/mL each; Duchefa). The medium was renewed every 2–3 days, and cells were passaged twice a week when

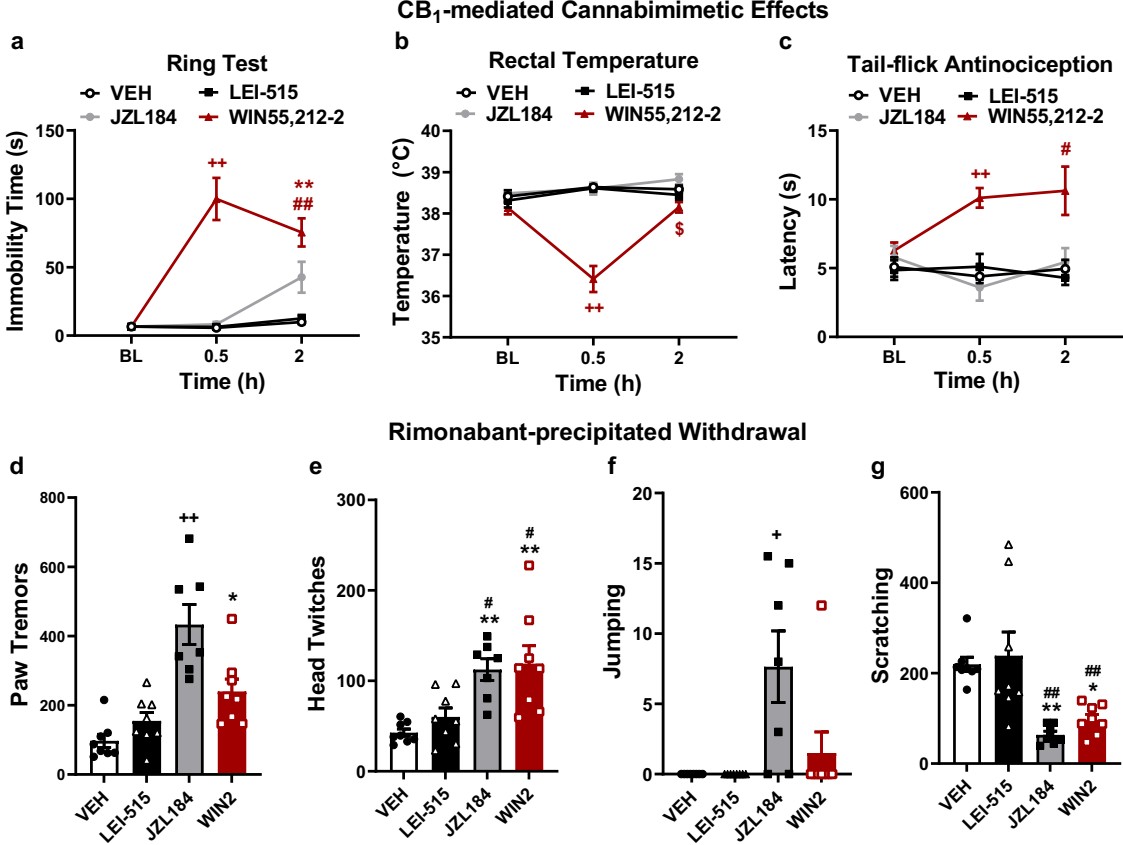

**Fig. 8 | The peripherally restricted MAGL inhibitor LEI-515 does not induce CB$_1$-dependent centrally mediated side effects or physical dependence. a–c** WIN55,212-2 (3 mg/kg i.p.) **a** increased immobility time in the ring test, **b** decreased body temperature and **c** increased tail-flick antinociception whereas LEI-515 (10 mg/kg i.p.) was inactive in all assays ($p > 0.8879$ vs. vehicle) $^{++}p < 0.01$ vs. all other groups; $^{**}p < 0.01$ vs. vehicle; $^{##}p < 0.01$, $^{#}p < 0.05$ vs. LEI-515; $^{\$}p < 0.05$ vs. JZL184. JZL184 ($n = 7$ male mice), all other groups ($n = 8$ male mice). **d–g** Rimonabant (10 mg/kg i.p.) precipitated signs of physical dependence including **d** paw tremors, **e** head twitches, **f** withdrawal jumps, and **g** decreased scratching bouts in mice

treated chronically with the global MAGL inhibitor JZL184 (16 mg/kg/day i.p. × 20 days) or the orthosteric cannabinoid agonist WIN55,212-2 (3 mg/kg/day i.p. × 20 days) but not in mice treated chronically with LEI-515 (10 mg/kg/day i.p. × 20 days) ($p > 0.6554$ vs. vehicle for all assays). $^{++}p < 0.01$ vs. all groups; $^{**}p < 0.01$, $^{*}p < 0.05$ vs. vehicle; $^{##}p < 0.01$, $^{#}p < 0.05$ vs. LEI-515. JZL184 ($n = 7$ male mice), all other groups ($n = 8$ male mice). All data are represented as mean values ± SEM. Statistical analysis: Two-way ANOVA with Tukey's multiple comparison test (**a–c**) or one-way ANOVA with Tukey's multiple comparison test (**d–g**). Source data and statistical analyses are provided as a Source Data file.

reaching 80–90% confluence. A day before transfection, HEK293T cells were relocated from confluent 10 cm dishes to 15 cm dishes. Preceding transfection, the medium was refreshed (13 mL). A blend of polyethyleneimine (PEI; 60 μg/dish) and plasmid DNA (20 μg/dish) in a 3:1 ratio was prepared in serum-free medium (2 mL) and allowed to incubate for 15 min at room temperature. Subsequently, the mixture was gently added dropwise to the cells, following which the cells were allowed to reach confluence over a 72-h period. Cells were then gathered by suspension in PBS, followed by centrifugation (200×*g*, 5 min). The resultant cell pellets were flash-frozen in liquid nitrogen and stored at −80 °C.

Cell pellets were thawed on ice and resuspended in lysis buffer A (20 mM HEPES (pH 7.2), 2 mM DTT, 250 mM sucrose, 1 mM MgCl$_2$, and 25 U/mL benzonase). Suspensions were homogenized by polytron (3 × 7 s, 20,000 rpm, SilentCrusher S; Heidolph, Schwabach, Germany), incubated on ice for 30 min, and subsequently centrifuged at 93,000×*g* for 30 min at 4 °C (Ti70 or Ti70.1 rotor; Beckman Coulter, Woerden, The Netherlands). The pellet was resuspended in storage buffer B (20 mM HEPES (pH 7.2), 2 mM DTT)]. Suspension was homogenized by polytron (1 × 10 s, 20,000 rpm). Protein concentrations were determined with Quick Start Bradford reagent (Bio-Rad, Hilversum, The Netherlands) or Qubit fluorometric quantitation (Life Technologies, Breda, The Netherlands). Membranes were diluted with storage buffer B to the

desired concentration, aliquoted, frozen in liquid nitrogen, and stored at −80 °C.

**Biochemical MAGL activity assays**

**MAGL activity assay (96-well format).** Biochemical MAGL activity assays were conducted in a 96-well format. The assays were carried out in HEMNB buffer (50 mM HEPES pH 7.4, 1 mM EDTA, 5 mM MgCl2, 100 mM NaCl, 0.5% (w/w) BSA) using black, flat-bottom 96-well plates (Greiner). Inhibitors were added from a 40× concentrated stock solution in DMSO. MAGL-overexpressing membrane preparations (0.3 μg per well) were incubated with the inhibitor for 20 min at room temperature (RT) in a total volume of 100 μL. Subsequently, a 100 μL assay mix containing glycerol kinase (GK), glycerol-3-phosphate oxidase (GPO), horse radish peroxidase (HRP), adenosine triphosphate (ATP), Amplifu™Red, and 2-arachidonoylglycerol (2-AG) was added. Fluorescence ($\lambda_{ex} = 535$ nm, $\lambda_{em} = 595$ nm) was measured at RT in 5 min intervals for 60 min on a GENios (Tecan) or Clariostar (BMG Labtech) plate reader. Final assay concentrations included 1.5 ng/μL MAGL-overexpressing membranes, 0.2 U/mL GK, GPO, and HRP, 125 μM ATP, 10 μM Amplifu™Red, 25 μM 2-AG, 5% DMSO, 0.5% ACN in a total volume of 200 μL. For KM and IC50 determinations, the assay was performed with variable 2-AG and inhibitor concentrations.

All measurements were conducted in $N = 2$ (individual plates), $n = 2$ (technical replicates on the same plate), or $N = 2$, $n = 4$ for

controls. Fluorescence values were corrected for the average fluorescence of the negative control (mock-membranes + vehicle). Slopes of the corrected data were determined in the linear interval. The $Z'$-factor for each assay plate was calculated using the formula $Z' = 1 - 3(\sigma pc + \sigma nc)/(\mu pc - \mu nc)$, and plates with $Z' \geq 0.6$ were accepted for further analysis (with $\sigma$ = standard deviation, $\mu$ = mean, pc = positive control and nc = negative control). For $IC_{50}$ determination, slopes were normalized to the positive control and analysed in a non-linear dose-response analysis with variable slope (GraphPad Prism 5.0).

For high-throughput screening (1536-well format, Corning 3724), the assays were performed as described for the 96-well format, with HEMNB buffer supplemented with 0.03% (w/w) Tween-20. Inhibitors (200x concentrated stock solution in DMSO, 20 nL per well) were added using acoustic dispensing (Labcyte 555 Echo Liquid Handler) and diluted in assay buffer (1 μL). Membranes (2 μL) were added, and mixtures were incubated for 30 min at RT, followed by the addition of assay mix (1 μL). Fluorescence endpoint measurement was performed after 45 min incubation at RT on an EnVision Multimode plate reader (Perkin Elmer). Fluorescence values were corrected for the average fluorescence of the negative control (MAGL-overexpressing membranes + 10 μM JZL184). Deselection assays were performed in a similar fashion but with glycerol (12.5 μM) instead of 2-AG as substrate. Further details regarding the HTS can be found in Supplementary Tables 1 and 2.

**Hit validation by activity-based protein profiling.** Hit validation by activity-based protein profiling involved adding inhibitor solutions (50 nL in DMSO, final concentration 10 μM) to 384-well plates using acoustic dispensing (Labcyte 555 Echo Liquid Handler). Mouse brain membrane preparation (10 μL, 2 mg/mL) was added, and the mixture was incubated for 30 min at RT. This was followed by incubation with FP-TAMRA (0.5 μL in DMSO, final concentration 100 nM, 10 min, RT Reactions were quenched with 4× Laemmli buffer (3.5 μL, final concentrations 60 mM Tris pH 6.8, 2% (w/v) SDS, 10% (v/v) glycerol, 5% (v/v) β-mercaptoethanol, 0.01% (v/v) bromophenol blue) for 30 min at RT. Samples (18;μg protein) were resolved by SDS–PAGE on a 10% polyacrylamide gel (180 V, 75 min). Gels were scanned using Cy3 channel settings (605/50 filter; ChemiDoc™ MP System, Bio-Rad). Fluorescence intensity was corrected for protein loading determined by Coomassie Brilliant Blue R-250 staining and quantified with Image Lab (Bio-Rad).

**Crystallization studies**
Human MAGL protein with mutations introduced at positions Lys36Ala, Leu169Ser and Leu176Ser[51] was produced by Cepter Biopartners LLC (Nutley, USA). A DNA fragment spanning the human His6-TEV-MAGL (K36A, L169S, L176S) was synthesized (GenScript) and inserted into pET28-using NcoI/EcoRI restriction sites, then transformed into *Escherichia coli* BL21(DE3) (Novagen). Cultures were inoculated and maintained in logarithmic growth at 37 °C in an LB medium containing appropriate antibiotics. When the $A_{600}$ of cultures reached 0.6–0.8, the cultures were chilled on ice, then expression was induced with 1 mM IPTG for 16 h at 23 °C. Cells were harvested by centrifugation and stored at −80 °C. The cell pellet was suspended in Buffer A [50 mM Tris–HCl (pH 7.5), 500 mM NaCl, 10% glycerol] with Benzonase (Sigma, 1:20000 dil.). The cells were disrupted by a microfluidizer, and insoluble material was removed by centrifugation at 40,000 rpm for 45 min (Beckman, 45Ti). The soluble lysate was filtered through a Steritop™ 0.22 μm PES membrane (Millipore), then loaded onto a HisTrap FF (GE Healthcare, 5 mL equilibrated in Buffer A. After washing with 25 mM imidazole in Buffer A, His6-TEV-MAGL(K36A, L169S, L176S) was eluted with 25–500 mM imidazole in Buffer A. Peak fractions containing His6-TEV-MAGL(K36A, L169S, L176S) were pooled and incubated with His6-TEV protease for overnight with dialysis against 2 L of Buffer B [50 mM Tris–HCl (pH 7.5),

200 mM NaCl, 5% glycerol, 2 mM DTT]. In order to remove the cleaved His6-tag and His6-TEV protease, the mixture was re-loaded onto HisTrap FF, and the flowthrough fraction containing tag-cleaved MAGL was collected. The target protein was concentrated to an appropriate volume using Amicon Ultra-15 (Millipore, 10 kDa MWCO) and loaded onto a HiLoad Superdex 75 26/60 (GE Healthcare), which was equilibrated in Buffer C [50 mM HEPES (pH 7.5), 200 mM NaCl, 2 mM DTT, 2 mM EDTA, 2% glycerol]. MAGL-containing fractions were pooled. A typical yield of tag-free hMAGL (K36A, L169S, L176S) was around 2.5–3.0 mg per 1 L LB culture. For crystallization the protein was concentrated to 10.8 mg/mL. Crystallization trials were performed in sitting drop vapour diffusion setups at 21 °C. Crystals appeared within 2 days out of 0.1 M MES pH 6.5, 6–13% PEG MME5K, 12% isopropanol. The structure of hMAGL (K36A, L169S, L176S) with LEI-515 was obtained by soaking the crystals for 16 h in crystallization solution supplemented with 10 mM LEI-515 dissolved in DMSO.

For data collection crystals were flash cooled at 100 K with 20% ethylene glycol added as cryo-protectant. X-ray diffraction data were collected at a wavelength of 1.00009 Å using an Eiger2X 16M detector at the beamline X10SA of the Swiss Light Source (Villigen, Switzerland). Data have been processed with XDS[52] and scaled with SADABS (BRUKER). The crystals belong to space group C2221 with cell axes of $a$ = 91.62 Å, $b$ = 127.57 Å, $c$ = 60.36 Å and diffract to a resolution of 1.55 Å. The structure was determined by molecular replacement with PHASER 2.8.3[53] using the coordinates of PDB entry 3pe6 as search model[51]. Difference electron density was used to place LEI-515. The structure was refined with programmes from the CCP4 suite 7.1.018[54]. Manual rebuilding was done with COOT 0.9.6[55]. Data collection and refinement statistics are summarized in Supplementary Table 3. The co-crystal structure of MAGL and LEI-515 is available in the PDB using code 8AQF.

**Selectivity assessment**
**Preparation of mouse tissue proteome.** Mouse tissues were isolated according to guidelines approved by the ethical committee of Leiden University. Isolated tissues were thawed on ice, dounce homogenized in lysis buffer A (20 mM HEPES pH 7.2, 2 mM DTT, 1 mM MgCl₂, 25 U/mL Benzonase), and incubated for 15 min on ice. Then, debris was removed by low-speed spin (2500×$g$, 1 min, 4 °C) and the supernatant was subjected to ultracentrifugation (100,000×$g$, 45 min, 4 °C, Beckman Coulter, Type Ti70 rotor) to yield the membrane fraction as a pellet and the cytosolic fraction in the supernatant. The membrane fraction was resuspended in lysis buffer B (20 mM HEPES pH 7.2, 2 mM DTT). The total protein concentration was determined with Quick Start Bradford assay. The obtained membranes and supernatant were flash-frozen in liquid nitrogen and stored in small aliquots at −80 °C until use.

**Activity-based protein profiling on mouse brain proteome.** To 19 μL mouse brain proteome (2 mg/mL) was added 0.5 μL of the inhibitor or pure DMSO, vortexed gently and incubated for 30 min at RT. Subsequently, 0.5 μL of either FP-TAMRA (final concentration: 100 nM) or MB064 (final concentration: 250 nm) was added to the proteome sample, vortexed gently and incubated for 10 min. The reaction was quenched by adding 10 μL of 4x Laemmli-buffer (240 mM Tris (pH 6.8), 8% (w/v) SDS, 40% (v/v) glycerol, 5% (v/v) β-mercaptoethanol, 0.04% (v/v) bromophenol blue) and 10 μL of quenched reaction mixture was resolved on 10% acrylamide SDS–PAGE (180 V, 75 min). Fluorescence was measured using a Biorad ChemiDoc MP system. Gels were then stained using Coomassie staining and imaged for protein loading control.

**Chemical proteomics.** Mouse tissue proteome (245 μL, 2.0 mg/mL) membrane or soluble fraction was incubated with vehicle (DMSO) or inhibitor (1 μM) in DMSO for 30 min at 37 °C. The proteome was

labelled with a probe cocktail (2.5 μM MB108 and 5 μM FP-Biotin, 30 min, 37 °C). The protein content was isolated by methanol/chloroform precipitation. Precipitation was done by the addition of 250 μL water, 666 μL MeOH and 166 μL chloroform with briefly vortexing after MeOH and chloroform addition. The precipitated protein was pelleted by centrifugation (10 min, 1500×g, RT). The upper and lower layers were removed without disturbing the floating pellet. Thereafter, the pellet was resuspended in 600 μL of MeOH using a probe sonicator (10 s, 20% amplitude). The protein was pelleted again by centrifugation (5 min, 18,400×g, RT) and the supernatant was removed. The protein pellet was resuspended in 250 μL urea buffer (250 μL, 6 M urea, 250 mM $NH_4HCO_3$), DTT (2.5 μL, final concentration: 10 mM) was added, vortexed briefly, spinned down briefly (10 s, 600×g, RT) and incubated for 15 min at 65 °C while shaking (600 rpm). After the samples were cooled down to RT, iodoacetamide (20 μL, final concentration: 40 mM) was added, vortexed briefly and incubated for 30 min at RT in the dark. Subsequently, SDS (70 μL, 10%) was added, vortexed briefly and incubated for 5 min at 65 °C. For 24 samples, 2.4 mL of avidin agarose beads (Thermo Scientific, 20219) were divided over three 15 mL tubes and washed with PBS (3 × 10 mL). The beads in each tube were resuspended in PBS (6 mL) and divided over 24 tubes (1 mL each). To each tube was added the denatured sample and PBS (2 mL) and the tubes were rotated with an overhead shaker at room temperature for 3 h. After centrifugation (2 min, 2500×g) and removal of the supernatant, the beads were consecutively washed with 0.5% SDS in PBS (w/v, 6 mL) and PBS (3 × 6 mL), each time centrifuging (2 min, 2500×g). The beads were transferred to a 1.5 mL low binding epp (Sarstedt, 72.706.600) with on-bead digestion buffer (250 μL, 100 mM Tris pH 8.0, 100 mM NaCl, 1 mM $CaCl_2$, 2% v/v acetonitrile) and to each sample was added 1 μL trypsin solution (0.5 μg/μL trypsin (Promega, V5111), 0.1 mM HCl). Proteins were digested at 37 °C with shaking (950 rpm) overnight. Formic acid (12.5 μL) was added to each sample and the beads were filtered off using a biospin column (Bio-Rad, 7326204), the flow-through was collected in a 2 mL epp. Samples were purified using StageTips. Each StageTip was conditioned with MeOH (50 μL, centrifugation: 2 min, 300×g), followed by StageTip solution B (50 μL, 80% v/v acetonitrile, 0.5% v/v formic acid in MilliQ, centrifugation for 2 min, 300×g) and StageTip solution A (50 μL, 0.5% v/v formic acid in MilliQ, centrifugation for 2 min, 300×g). Next, the samples were loaded, centrifuged (2 min, 600×g) and the peptides on the StageTip were washed with StageTip solution A (100 μL, centrifugation for 2 min, 600×g). The StageTips were transferred to a new 1.5 mL low binding epp and the peptides were eluted with StageTip solution B (100 μL, centrifugation for 2 min, 600×g). The solvents were evaporated to dryness in a SpeedVac concentrator at 45 °C for 3 h. Samples were reconstituted in LCMS solution (50 μL, 3% v/v acetonitrile, 0.1% v/v formic acid, 1 μM yeast enolase peptide digest (Waters, 186002325) in MilliQ)[56].

The peptides were separated on an UltiMate 3000 RSLCnano system set in a trap-elute configuration with a nanoEase M/Z Symmetry C18 100 Å, 5 μm, 180 μm × 20 mm (Waters) trap column for peptide loading/retention and nanoEase M/Z HSS C18 T3 100 Å, 1.8 μm, 75 μm × 250 mm (Waters) analytical column for peptide separation. The column was kept at 40 °C in a column oven. Samples (5 μL/sample) were injected on the trap column at a flow rate of 15 μL/min for 2 min with 99% mobile phase A (0.1% FA in ULC-MS grade water (Biosolve)), 1% mobile phase B (0.1% FA in ULC-MS grade acetonitrile (Biosolve)) eluent. The 85 min LC method, using mobile phase A and mobile phase B controlled by a flow sensor at 0.3 μL/min with average pressure of 400–500 bar (5500–7000 psi), was programmed as gradient with linear increment to 1% B from 0 to 2 min, 5% B at 5 min, 22% B at 55 min, 40% B at 64 min, 90% B at 65–74 min and 1% B at 75–85 min. The eluent was introduced by electro-spray ionization (ESI) via the nanoESI source (Thermo) using stainless steel Nano-bore emitters (40 mm, OD 1/32", ES542, Thermo Scientific). The QExactive HF was operated in positive mode with data-dependent acquisition without the use of lock mass, default charge of 2+ and external calibration with LTQ Velos ESI positive ion calibration solution (88323, Pierce, Thermo) every 5 days to <2 ppm. The tune file for the survey scan was set to a scan range of 350–1400 m/z, 120,000 resolution (m/z 200), 1 microscan, automatic gain control (AGC) of 3e6, max injection time of 100 ms, no sheath, aux or sweep gas, spray voltage ranging from 1.7 to 3.0 kV, capillary temp of 250 °C and an S-lens value of 80.

Data was analysed with MaxQuant 2.0 and GraphPad Prism 9.0. The mass spectrometry proteomics data generated have been deposited to the ProteomeXchange Consortium via the PRIDE partner repository with the dataset identifier PXD042220.

**Competitive ABPP on hHSL and mHSL overexpression cell lysates.** To 19 μL overexpression lysate (2 mg/mL) was added 0.5 μL of the inhibitor or pure DMSO, vortexed gently and incubated for 30 min at RT. Subsequently, 0.5 μL FP-TAMRA was added to the proteome sample, vortexed gently and incubated for 10 min. The reaction was quenched by adding 10 μL of 4× Laemmli-buffer (240 mM Tris (pH 6.8), 8% (w/v) SDS, 40% (v/v) glycerol, 5% (v/v) β-mercaptoethanol, 0.04% (v/v) bromophenol blue) and 10 μL of quenched reaction mixture was resolved on 10% acrylamide SDS–PAGE (180 V, 75 min). In-gel fluorescence was detected in the Cy3-channel on a ChemiDoc MP imaging system (Bio-Rad). The gel was stained using Coomassie G250 blue staining as a protein loading control. Fluorescence and Coomassie intensities were quantified using ImageLab 6.0 software (Bio-Rad) and normalized log-transformed data was fitted using the variable slope dose–response function in GraphPad Prism 9.0.

## Cellular target engagement

**Cell culture.** U-87 MG (human glioblastoma) cells were obtained from ATCC (catalogue number: HBT-14) and were cultured in DMEM (Sigma, D6546), supplemented with 10% foetal calf serum (FCS), stable glutamine (glutaMAX) and penicillin and streptomycin (100 μg/mL), at 37 °C and 7% $CO_2$ in a humidified incubator. Cells were passaged through detachment with trypsin/EDTA in PBS for 3 min at RT, followed by the addition of fresh medium. After cell counting, cells were seeded in a fresh medium at the appropriate confluency. Cell lines were regularly tested for mycoplasma contamination and consistently tested negative. Cultures were discarded after 2–3 months of use.

**In situ treatments.** The term in situ is used to describe experiments in which live cell cultures are treated with inhibitors. U-87 MG cells were seeded at 200,000 cells/well in six-well plates, 24 h prior to treatment. Culture medium was aspirated and replaced with medium (DMEM supplemented with 2% FCS, glutaMAX, P/S) containing vehicle (0.1% DMSO) or inhibitor (final concentrations: 1 nM–10 μM) was added. After incubation for 1 h, the MAGL-specific fluorescent activity-based probe LEI463-Cy5[33], was added as a 6× concentrated stock in medium (final concentration 10 nM). After incubation for an additional 1 h, the medium was aspirated and the cells were harvested by resuspension in cold PBS and pelleted (1000×g, 2 min, RT). PBS was removed and the cell pellets were flash-frozen in liquid nitrogen and stored at −80 °C until further use.

**Cell lysate preparation and gel-based ABPP: in situ treatment.** Cell pellets were thawed on ice, lysed in SDS-lysis buffer (50 mM Tris buffer pH 7.0, 2 mM EDTA, 1% SDS) and sonicated (3 × 10 s, amplitude 25%) in a Q700 sonicator (QSonica). Protein concentrations were determined using Pierce BCA protein assay kit (Thermofisher). 4x Laemmli-buffer (240 mM Tris (pH 6.8), 8% (w/v) SDS, 40% (v/v) glycerol, 5% (v/v) β-mercaptoethanol, 0.04% (v/v) bromophenol blue) was added to the samples and 10 μg of protein was resolved by SDS–PAGE (10% 29:1 acrylamide:bisacrylamide). In-gel fluorescence was detected in the Cy5-channel (700/50 nm) on a ChemiDoc MP imaging system

(Bio-Rad). The gel was stained using Coomassie G250 blue staining as a protein loading control. Fluorescence and Coomassie intensities were quantified using ImageLab 6.0 software (Bio-Rad) and normalized log-transformed data was fitted using the variable slope dose–response function in GraphPad Prism 9.0.

## Targeted lipidomics

**Lipidomics sample preparation.** Here, $3 \times 10^6$ HS578t cells, obtained from ATCC (catalogue number: HTB-126), were seeded 1 day before treatment in 10 cm dishes. Before treatment, cells were washed twice with warm PBS and then treated with vehicle or compound in 10 mL medium without serum. Washing with cold PBS (1×) followed by gathering in 1.5 mL Eppendorf tubes and centrifugation (10 min, 1500×$g$). PBS was removed and the cell pellets were flash-frozen with liquid nitrogen and stored at −80 °C. Next, 10% of each cell sample (collected during harvesting) was used to determine the protein concentration using a Bradford assay (Bio-Rad) for normalization after lipid measurements.

**Lipid extraction.** Lipid extraction was performed on ice. Samples were thawed on ice and spiked with 10 μL deuterium labelled internal standard mix, vortex and incubated for 5 min on ice. Subsequently, 100 μL 0.5% sodium chloride and 100 μL ammonium acetate buffer (0.1 M, pH 4) were added. After extraction with 1 mL methyl tert-butyl ether (MTBE), tubes were thoroughly mixed for 7 min using a bullet blender blue (Next advance Inc., Averill Park, NY, USA) at speed 8, followed by a centrifugation step (16,000×$g$, 11 min, 4 °C). Next, 925 μL of the upper MTBE layer was transferred into a clean 1.5 mL Safe-Lock Eppendorf tube. Samples were dried in a speedvac (Eppendorf, 45 min, 30 °C) and reconstituted in acetonitrile/water (30 μL, 90:10, v/v). The samples were thoroughly mixed for 4 min, followed by a centrifugation step (10,000×$g$, 4 min, 4 °C) and transferred to an LC–MS vial (9 mm, 1.5 mL, amber screw vial, KG 090188, Screening Devices) with insert (0.1 mL, tear drop with plastic spring, ME 060232, Screening devices). 5 μL of each sample was injected into the LC–MS/MS system.

**LC–MS/MS analysis.** A targeted analysis of 26 compounds, including endocannabinoids, related N-acylethanolamines (NAEs) and free fatty acids (Supplementary Table 6), was measured using an Acquity UPLC I class binary solvent manager pump in conjugation with a tandem quadrupole mass spectrometer as mass analyser (Waters Corporation, Milford, USA). The separation was performed with an Acquity HSS T3 column (2.1 × 100 mm, 1.8 μm) maintained at 45 °C. The aqueous mobile phase A consisted of 2 mM ammonium formate and 10 mM formic acid, and the organic mobile phase B was acetonitrile. The flow rate was set to 0.55 mL/min; initial gradient conditions were 55% B held for 0.5 min and linearly ramped to 60% B over 1.5 min. Then the gradient was linearly ramped to 100% over 5 min and held for 2 min; after 10 s the system returned to initial conditions and held 2 min before the next injection. Electrospray ionization-MS and a selective Multiple Reaction Mode (sMRM) was used for endocannabinoid quantification. Individually optimized MRM transitions using their synthetic standards for target compounds and internal standards are described in Supporting Information Table 6. Peak area integration was performed with MassLynx 4.1 software (Waters Corporation). The obtained peak areas of targets were corrected by appropriate internal standards peak area. Calculated response ratios, determined as the peak area ratios of the target analyte to the respective internal standard peak area, were used to obtain absolute concentrations from their respective calibration curves. Data were tested for significance with GraphPad 6.0.

## In vitro ADME profile and in vivo pharmacokinetics

**Microsomal clearance.** Pooled commercially available microsome preparations from male mouse microsomes. For humans, ultrapooled liver microsomes were purchased to account for the biological variance in vivo from human liver tissues. For the microsome incubations (incubation buffer 0.1 M phosphate buffer pH 7.4), 96-deep well plates were applied, which were incubated at 37 °C on a TECAN equipped with Te-Shake shakers and a warming device. The reduced nicotinamide adenine dinucleotide phosphate regenerating system consisted of 30 mM glucose-6-phosphate disodium salt hydrate; 10 mM NADP; 30 mM MgCl$_2$ × 6H$_2$O and 5 μg/μL glucose-6-phosphate dehydrogenase in 0.1 M potassium phosphate buffer pH 7.4. Incubations of LEI-515 at 1 μM in microsome incubations of 0.5 μg/μL plus cofactor reduced nicotinamide adenine dinucleotide phosphate were performed in 96-well plates at 37 °C. After 1, 3, 6, 9, 15, 25, 35 and 45 min 40 μL incubation solutions were transferred and quenched with 3:1 (v/v) acetonitrile containing internal standards. Samples were then cooled and centrifuged before analysis by LC–MS/MS. The log peak area ratios (test compound peak area/internal standard peak area) were plotted against incubation time using a linear fit. The calculated slope was used to determine the intrinsic clearance: CL$_{int}$ (μL/min/mg protein) = −slope(min$^{-1}$) × 1000/(protein concentration (μg/μL)).

**Plasma protein binding.** Pooled and frozen plasma from humans and mice were obtained from BioreclamationIVT. The Teflon equilibrium dialysis plate (96-well, 150 μL, half-cell capacity) and cellulose membranes (12–14 kDa molecular weight cut-off) were purchased from HT-Dialysis (Gales Ferry). Both biological matrix and phosphate buffer pH were adjusted to 7.4 on the day of the experiment. The determination of unbound compound was performed using a 96-well format equilibrium dialysis device with a molecular weight cut-off membrane of 12–14 kDa. The equilibrium dialysis device itself was made of Teflon to minimize nonspecific binding of the test substance. Compound were tested with an initial total concentration of 1000 nM, one of the cassette compounds being the positive control diazepam. Equal volumes of matrix samples containing substances and blank dialysis buffer (Soerensen buffer at pH 7.4) were loaded into the opposite compartments of each well. The dialysis block was sealed and kept for 5 h at a temperature of 37 °C and a 5% CO$_2$ environment in an incubator. After this time, equilibrium has been reached for most small molecule compounds with a molecular weight of <600. The seal was then removed and the matrix and buffer from each dialysis were prepared for analysis by LC–MS/MS. All protein binding determinations were performed in triplicate. The integrity of membranes was tested in the HT-Dialysis device by determining the unbound fraction values for the positive control diazepam in each well. At equilibrium, the unbound drug concentration in the biological matrix compartment of the equilibrium dialysis apparatus was the same as the concentration of the compound in the buffer compartment. Thus, the percentage unbound fraction ($f_u$) was calculated by determining the compound concentrations in the buffer and matrix compartments after dialysis as follows: %$f_u$ = 100 × buffer concentration after dialysis/matrix concentration after dialysis. The device recovery was checked by measuring the compound concentrations in the matrix before dialysis and calculating the percent recovery (mass balance). The recovery must be within 80–120% for data acceptance.

**Passive permeability in LLC-PK1 cells.** LLC-PK1 cells overexpressing human P-glycoprotein (encoded by the Multi-Drug Resistance (MDR)1 gene), were obtained from Dr. A. Schinkel, The Netherlands Cancer Institute (Amsterdam, The Netherlands) under licence agreement. Compounds dosed at 1 μM were evaluated for permeability in the apical-to-basolateral and basolateral-to-apical directions using a liquid handling robot (Tecan, Männedorf, Switzerland)[57]. Samples were collected from triplicate wells of donor and receiver compartments after a 3.5-hour incubation in the presence or absence of P-gp inhibitor (zosuquidar, 1 μM). Drug concentrations were measured by high-performance liquid chromatography–tandem mass spectrometry (LC–MS/MS). The extracellular marker lucifer yellow (10 μM) was also

added to ensure monolayer integrity; values exceeding 25 nm/s were eliminated from the analysis.

For the permeability data assessment, the following equation was used:

$$P_{\text{app}}(\text{nm/s}) = \frac{1}{A \times C0} \times \frac{dQ}{dt} \qquad (1)$$

where $P_{\text{app}}$, A, $C0$ and $dQ/dt$ represent the mean, bidirectional apparent permeability in the presence of P-gp inhibitor, the filter surface area, the initial concentration of test substances and the amount per time period (3.5 h), respectively.

**Pharmacokinetic analysis of LEI-515.** Test compounds were formulated according to respective protocols either by dissolution (intravenous, i.v.) or in a glass potter until homogeneity was achieved (oral, p.o.). Formulations were injected i.v. using a 30 G needle in the lateral tail vein mice yielding a 10 mg/kg dose. For p.o. applications, animals were gavaged (yielding a 10 mg/kg dose). At the following time points blood was drawn into EDTA vials: 0.083, 0.25, 0.5, 1, 2, 4, 7 and 24 h. Three animals each were used for the i.v. and p.o. arm. Animals were distributed randomly over the time course and at each time point, a volume of 100 μL of blood was taken. Quantitative plasma measurement of the compound was performed by LC–MS/MS analysis. Pharmacokinetic analysis was conducted using Phoenix WinNonlin v.6.4 software using a noncompartmental approach consistent with the route of administration. The maximum concentration ($C_{\text{max}}$), time to the maximum concentration ($T_{\text{max}}$) and area under the curve (AUC) were determined from the plasma concentration profiles. The bioavailability was calculated by dividing the AUC of the p.o. group by the AUC of the i.v. group. Parameters (CL, Vss, $t_{1/2}$) were estimated using nominal sampling times relative to the start of each administration.

**In vivo target engagement**

**Animals.** Nine to ten weeks old wild-type C57BL/6J male mice were purchased from Charles River France and stabilized, single-caged, in Type II cages for at least seven days after delivery. All mice were housed in a temperature- and humidity-controlled (23–25 °C, 35–65%) environment in a 12-h light/dark cycle. The animals had free access to water and food and were fed with a standard laboratory animal diet (pelleted Provimi Kilba cat. no. 3436). Animals are checked and inspected by trained animal caretakers daily. All animal experiments performed in this study were approved by the Cantonal Veterinary Office Basel-Stadt (animal welfare division), under the license nr.1902. Animals were euthanised with isoflurane anaesthesia (inhalation with 2–3 vol% isoflurane and 100% oxygen) and decapitated by using a guillotine.

**Experimental design.** In brief, six mice per group were administered vehicle, 30 or 100 mg/kg LEI-515 at a volume of 10 mL/kg (formulation: 5% Vitamin E-TGPS in water and sodium hydroxide, equilibrated at RT for 30 min) by oral gavage. The mice were euthanized 2 h after compound/vehicle administration and their left hemisphere, intestine, liver and lung tissues were prepared for LC–MS/MS quantification of analytes.

**Tissue collection.** The animals were euthanized 2 h after compound/vehicle administration, placed under isoflurane anaesthesia (inhalation with 2–3 vol% isoflurane and 100% oxygen) and decapitated. The brains were quickly removed from the skull, and the left and right hemispheres were dissected separately. The lungs, liver and colon were dissected. The colon was longitudinally opened, placed in ice-cold PBS and vortexed at max speed for 15 s. The washing step was repeated three times. All tissue samples were snap-frozen in liquid nitrogen and stored at −80 °C.

**LC–MS/MS sample preparation.** Tissue samples were placed in 7 mL Precellys tubes containing ceramic beads (Bertin Instruments, cat. no. P000935-LYSK0-A.0) and homogenized in methanol (1 mL methanol/100 mg brain tissue) three times for 10 s at 6000 rpm using a Precellys homogenator (Bertin Instruments, cat. no. P000062-PEVO0-A). Then 100 μL of the produced homogenate was centrifuged at 13,000 rpm for 10 min at 4 °C. For quantification, 50 μL of supernatant diluted 1:100 in methanol were transferred into a 96-well deep well plate, and 50 μL methanol containing internal standards (5 μM 2-AG-d5 and 10 μM AA-d8, purchased from Cayman) and 5 μL methanol were added. Calibration standards were prepared into the same 96-well deep well plate as follows: 50 μL pooled supernatant diluted 1:100 in methanol (i.e., pooled supernatant from all the brain samples), 50 μL methanol containing internal standards (5 μM 2-AG-d5 and 10 μM AA-d8) and 5 μL calibration standards. Analytes and internal standards stock solutions 10 mM were prepared in ethanol or acetonitrile and stored at −80 °C. Calibration standards stock solutions (starting concentrations 100 μM 2-AG, 200 μM AA) were prepared in methanol by 1:1 serial dilutions and stored at −80 °C.

**LC–MS/MS analysis.** Analyses were conducted on an LC–MS/MS system consisting of a Waters Xevo-TQ-S mass spectrometer connected to a complete Waters Acquity I-class ultra-performance liquid chromatography system with a flow-through needle sample manager. The auto-sampler was cooled down to 10 °C prior to analysis. A mixture of acetonitrile/methanol/water 40/40/20 (v/v/v) was used as a wash solvent. The injection volume of the sample was 2 μL. Compounds were separated on a Kinetex EVO C18 core–shell column (100 × 2.1 mm, 1.7 μM particle size, purchased from Phenomenex) using 0.1% acetic acid (solvent A) and acetonitrile (solvent B) as eluents at a flow rate of 0.3 mL/min and an oven temperature of 45 °C. The gradient was running from 30% B to 90% B within 5 min. The overall cycle time including column reconditioning was 7 min. Detection was done using a Waters Xevo-TQ-S mass spectrometer operating in positive and negative electrospray mode with both quadrupoles tuned to unit mass resolution. Nitrogen was used as both nebulization and desolvation gas with a nebulizer gas flow of 150 l/h and a desolvation gas flow of 800 l/h respectively and a temperature of 500 °C. Argon was used as a collision gas with a flow rate of 0.15 mL/min. Analyte and internal standards were measured in multiple reaction monitoring modes recording the mass transitions (Supplementary Table 7).

**Calibration and calculation.** Calibration was done in a brain matrix. The calibration range was linear up to 1000 nM for prostaglandins, up to 5–10 μM for 2-AG and AA. The calibration curves were fitted using linear regression function with $1/y$ weighting, excluding zero. Absolute concentrations were calculated in Excel 2016 using formula 2. Data were tested for significance with GraphPad 7.0

$$\text{Absolute analyte concentration} = \frac{\text{peak area analyte}/\text{peak area internal standard}}{\text{slope of calibration curve}}$$
$$(2)$$

**Acute liver injury model**

**Animals.** For the acute liver injury study, 64 mice were used (39 males and 25 females). Mice group-housed within a cage (3–4 mice/cage) received the same treatment. 40 mice were used for the liver injury study (CCl₄; Fig. 5) and 24 for controls (Supplementary Fig. 6). Animal usage and experimental protocol were in accordance with the guidelines of the National Institutes of Health, Institutional Animal Care and Use Committee of the National Institute on Alcohol Abuse and Alcoholism (Bethesda, MD). Eight to 12-week-old male and female C57BL/6J mice were used in the study (The Jackson Laboratory, Bar Harbor, Maine, strain#: 000664). All mice were housed in a temperature- and humidity-controlled (20.5–23.9 °C, 30–70%) environment in a 12-h

light/dark cycle. Animals were fed NIH 31 Rodent Diet (Harlan/Teklad Catalogue #7017) provided ad lib via stainless steel feeders that may or may not be integral to the wire bar lid. Animals are checked and inspected by trained animal caretakers daily. At the end of experimental procedures, mice were terminally anesthetized (isoflurane 5%) and exsanguinated followed by cervical dislocation.

**Experimental design.** A single dose of CCl₄ (10% in corn oil, 2 mL/kg; Fisher Scientific, Pittsburgh, PA, USA) was administered intraperitoneally and mice were sacrificed by exsanguination 24 h after the induction of acute liver injury[58]. Based on our previous studies with JZL184 in severe acute liver injury models[9] we selected comparable doses of LEI-515 or its vehicle and the same route of administration. LEI-515 (at 10 and 30 mg/kg doses) or its vehicle was administered intraperitoneally to mice 30 min prior to CCl₄ treatment.

**Determination of liver necrosis.** Serum activities of alanine aminotransferase (ALT) and aspartate aminotransferase (AST), markers of liver injury/necrosis, were measured immediately after collection by using a colorimetric kit (Fisher Scientific).

**Histology.** Liver samples were fixed in 10% buffered formalin (Fisher Scientific) and then processed in paraffin for histology. Tissue sections (5 μm) were stained with hematoxylin and eosin (H&E) for histological evaluation of liver injury by CCl₄. Immunostaining of CD45 was performed as described earlier to determine tissue leucocyte infiltration[59]. Immunolabeling of 4-hydroxinonenal was also performed to assess hepatic oxidative stress following CCl₄ treatment. Sections were visualized and images were acquired using BX-41 microscope (Olympus, Tokyo, Japan). The extent of liver necrosis, levels of 4HNE adducts, as well as the determination of CD45 cell count, were quantified on 6–8 random areas of liver sections. Images were analysed by using ImageJ 1.53 software (National Institutes of Health, Bethesda, MD, USA). Data were tested for significance with GraphPad 9.0.

### In vivo assessment of LEI-515

**Animals.** Male (n = 110) and female (n = 31) C57Bl6/J mice (Jackson Labs, Bar Harbor, ME) 12 weeks of age at the start of each experiment were used in behavioural studies. All mice were group housed in a temperature- and humidity-controlled (21–25 °C, 30–70%) environment in a 12-h light/dark cycle. Food (Teklad global diets #2918) and water were provided ad libitum. Animals were monitored and inspected by trained lab animal resources staff daily. In all studies, mice were randomized to experimental conditions and the experimenter was blinded to the pharmacological treatment. Males and female mice were tested in separate cohorts. All experimental procedures were approved by the Bloomington Institutional Animal Care and Use Committee of Indiana University, complied with all ethical guidelines of the National Institutes of Health (NIH) Guide for Care and Use of Laboratory Animals and followed guidelines outlined by the International Association for the Study of Pain.

**Drugs and chemicals.** Paclitaxel (Tecoland Corporation, Irvine, CA) was dissolved in a vehicle containing cremophor EL (Sigma Life Science, St. Louis, MO), ethanol (Decon Laboratories, King of Prussia, PA), and saline (Baxter Healthcare Corporation, Deerfield, IL) in a 1:1:18 ratio as previously described[36–40]. LEI-515 was synthesized by the van der Stelt lab (University of Leiden, Netherlands) and dissolved in a vehicle containing dimethyl sulfoxide (DMSO) (Sigma Life Science, St. Louis, MO):cremophor EL:saline in a 1:1:18 ratio for i.p. administration. LEI-515 was dissolved in a vehicle containing DMSO: Kolliphore (Sigma Life science, St. Louis, MO): 5% mannitol in water (Rigaku Reagents, Bainbridge Island, WA) in a 1:1:8 ratio for oral administration. AM251 (Cayman Chemical, Ann Arbor, MI), AM630 (Cayman Chemical, Ann Arbor, MI), AM6545 (provided by Alex Makriyannis, Center for Drug

Discovery, Northeastern University, Boston, MA), SR144528 (NIDA Drug Supply Programme, Research Triangle Institute, Research Triangle Park, NC) and Rimonabant (NIDA Drug Supply Programme, Research Triangle Institute, Research Triangle Park, NC) were dissolved in a vehicle containing DMSO:ethanol:ALKAMULS EL-620 (Solvay, Brussels, Belgium):saline in a 5:2:2:16 ratio. JZL184 (Cayman Chemical, Ann Arbor, MI) and WIN55,212-2 (Sigma Life Science, St. Louis, MO) were dissolved in DMSO:cremophor EL:saline solution in a 1:1:18 ratio to match LEI-515 i.p. vehicle. Drugs were administered in a volume of 10 mL/kg except where noted.

**Assessment of mechanical paw withdrawal thresholds.** Mechanical paw withdrawal thresholds (measured in grams (g)) were assessed using an electronic von Frey aesthesiometer in conjunction with a 90-g probe (model Almemo 2390-5; IITC, Woodland Hills, CA) as described previously by our groups[36–40]. Mice were placed on an elevated metal mesh table and were habituated to inverted plastic cages placed on the testing platform for at least 30 min to reduce exploratory behaviour prior to behavioural testing; this handling protocol was consistent across all testing days. After habituation, paw withdrawal thresholds were measured in duplicate per paw with the midplantar region of each hind paw being stimulated by a semiflexible tip connected to the aesthesiometer, increasing force until withdrawal response (paw retraction) was observed. Mechanical force was then terminated, and the force (in grams, g) to produce paw withdrawal was recorded. Mechanical paw withdrawal thresholds are reported as the mean of the duplicate determinations in each hind paw, averaged across animals. Paw withdrawal thresholds were measured before and at various timepoints following paclitaxel injections (i.e. day 0, 4, 8, 12 and 16 after initiation of paclitaxel dosing) and following pharmacological manipulations.

**Chemotherapy-induced neuropathic nociception.** Paclitaxel was administered at a dose of 4 mg/kg i.p. once daily on alternate days (i.e. day 0, 2, 4, and 6; cumulative dose: 16 mg/kg i.p.) in a volume of 6.67 mL/kg for a total of four injections as previously described by our groups[36–40]. Control mice received an equal volume of cremophor-based vehicles. Development of paclitaxel-induced mechanical hypersensitivity was assessed on days 0, 4, 8, and 12 following initiation of paclitaxel dosing prior to initiation of pharmacological manipulations in all studies.

**Pharmacological manipulations.** Separate studies examined the impact of either oral or i.p. dosing with LEI-515 or vehicle in mice of both sexes. Dose–response of LEI-515 in suppressing paclitaxel-induced mechanical hypersensitivity was assessed during the maintenance phase of paclitaxel-induced hypersensitivity when neuropathy was stable. Separate groups always received vehicles at the same times and were tested under identical conditions. Doses were administered once daily across 6 consecutive days (i.e., day 12–17 post initiation of paclitaxel dosing) using a within-subjects dosing paradigm for each route of administration. In chronic oral dosing studies, mechanical paw withdrawal thresholds were tested on each day before and 2 h after administration of LEI-515 or vehicle. In chronic i.p. dosing studies, mechanical paw withdrawal thresholds were tested before and 30 min post-injection of LEI-515 or vehicle.

Duration of action was assessed in the same mice used for the dose–response study after a washout period of 72 h; mechanical paw withdrawal thresholds were always reassessed before administration of LEI-515 or vehicle to ensure paclitaxel-induced hypersensitivity was fully intact. Mice were administered LEI-515 (10 mg/kg) or vehicle (10 mL/kg) via oral gavage, and mechanical paw withdrawal thresholds were assessed at 0.5, 2, 6, 24, and 72 h post LEI-515 or vehicle (p.o.), respectively. Separate groups of mice received LEI-515 (10 mg/kg) or vehicle (10 mL/kg) via i.p. injection, and mechanical paw withdrawal

thresholds were assessed at or at 0.5, 2, 6, and 24 h post LEI-515 or vehicle (i.p.). After a 14-day washout period, chronic dosing studies were performed for each route of drug administration. Mice received either LEI-515 (10 mg/kg) or vehicle (10 mL/kg) for 10 consecutive days (i.p. or p.o.). Mechanical paw withdrawal thresholds were tested on days 1, 4, 7, and 10 of repeated dosing and before each injection on each respective test day. Mechanical paw withdrawal thresholds were reassessed 2 h post-oral administration of LEI-515 or vehicle. Mechanical paw withdrawal thresholds were reassessed 30 min post i.p. administration of LEI-515 or vehicle. Mechanical paw withdrawal thresholds were also measured over 10 days following termination of chronic dosing (i.e., days 11, 14, 17, and 20 of chronic dosing) to assess whether antinociceptive efficacy of LEI-515 was sustained after cessation of repeated dosing.

In a separate study, the pharmacological specificity of the anti-allodynic effects of LEI-515 was assessed using paclitaxel-treated mice in the maintenance phase of behavioural hypersensitivity (i.e., 16 days after initial paclitaxel treatment). On day 16, baseline mechanical thresholds after paclitaxel were assessed. Then, mice received a pre-treatment (i.p.) with either AM251 (5 mg/kg), AM6545 (10 mg/kg), AM630 (5 mg/kg), or SR144528 (2.1 mg/kg) or vehicle (DMSO; ethanol: ALKAMULS EL-620: saline in a 5:2:2:16 ratio) in separate groups of mice. Mechanical paw withdrawal thresholds were then reassessed to assess the impact of each antagonist treatment on mechanical paw withdrawal thresholds prior to administration of LEI-515 or its vehicle. Then, LEI-515 (10 mg/kg) or vehicle (10 mL/kg) was administered (p.o.), and mechanical paw withdrawal thresholds were reassessed 0.5, 2, 6, and 24 h post administration of LEI-515/vehicle. A separate group of mice were treated with the cremophor vehicle in lieu of paclitaxel and was used to assess the impact of oral dosing with LEI-515 or vehicle (p.o.) on mechanical paw withdrawal thresholds in the absence of chemotherapy treatment; consequently, the experimenter was blinded to the pharmacological treatment (vehicle vs. LEI-515) but not to the chemotherapy condition (i.e. cremophor vehicle vs. paclitaxel).

ED$_{50}$ values for the suppression of paclitaxel-induced mechanical hypersensitivity were generated for each sex for each route of administration; paw withdrawal thresholds (in g) were converted to % maximal possible antinociceptive effect (%MPE) using the formula: (experimental value−post-paclitaxel baseline)/(pre-paclitaxel baseline−post-paclitaxel baseline)×100. ED$_{50}$ and 95% confidence intervals were calculated based on fitting normalized log-transformed data using the log agonist vs. response (three-parameter) least squares fit function in GraphPad Prism 9.0. All other analyses in paclitaxel-treated mice were performed on paw withdrawal thresholds in g.

**Cannabinoid triad for assessing cardinal signs of CB$_1$ activation.** The ability of LEI-515 to produce cardinal signs of CB$_1$ activation (i.e. catalepsy, hypothermia, and tail-flick antinociception) was measured in male mice as described previously by our group and others[36,60]. Mice were habituated to the testing room for 30 min in their home cages prior to behavioural testing. Baseline measures were obtained for each dependent measure prior to pharmacological manipulations. Subsequently, separate groups of male mice received a single (i.p.) injection of LEI-515 (10 mg/kg i.p.), JZL184 (16 mg/kg i.p.), WIN55,212-2 (3 mg/kg i.p.) or vehicle. Then, responses were reassessed at 0.5 and 2 h following injection in the following order: ring test, rectal temperature, and tail-flick antinociception.

Ring test: The time spent immobile (minus respiratory movements) on an elevated wire ring (6.35 cm diameter wire ring suspended 16 cm above a horizontal platform) over a 5 min observation interval was measured to assess catalepsy.

*Rectal temperature*: Body temperature (°C) was measured using a thermometer (Phytotomy Instruments, Inc., Clifton, NJ) attached to a rectal probe (Braintree Laboratories, Inc., Braintree, MA).

*Tail-flick antinociception*: The hot water tail-immersion test was conducted by submerging the distal 2 cm of the tail of a mouse in a hot water bath (52 °C) and recording the latency to remove the tail/elicit a 'flick' response. Prior to injection, three baseline values were obtained with a 10 min inter-trial interval and a maximum cut-off latency of 20 s to avoid tissue damage.

**CB$_1$-dependent withdrawal.** Male mice were challenged with the CB1 antagonist rimonabant to assess CB$_1$ receptor-dependent cannabinoid withdrawal. First, separate cohorts of mice received repeated once-daily injections (i.p.) with the vehicle, LEI-515, JZL184, or WIN55,212-2 (i.p.) for 20 consecutive days. On the final injection day, mice were placed underneath plexiglass chambers positioned on an elevated glass table and allowed to habituate to the apparatus for 30 min. Mirrors were placed underneath the glass table to facilitate observation of each animal's behaviour. Two hours following the final injection of the drug or vehicle, all groups were challenged with Rimonabant (10 mg/kg in a volume of 5 mL/kg i.p.), and behaviour was recorded for one hour using Logitech (Lausanne, Switzerland) C920 HD Pro 1080p cameras. Scoring was performed by an experimenter blinded to the experimental condition; each video was scored twice by the same rater to ensure inter-rater reliability and averaged into a single determination per dependent measure per subject.

Paw tremors, head twitches, scratching behaviour, and jumping behaviour were scored using BORIS (v 7.13.6) open source software. Paw tremors were counted as either individual or dual paw tremors (forelimb paws). Scratching behaviour was counted as either hind paw or forepaw scratches. Jumping behaviour was counted as all four paws leaving the table. Head twitches were counted as individual head-bobbing movements separate from walking or turning behaviour. Data from each mouse reflect the mean of both independent ratings of each dependent measure.

### Statistical analysis

Results are expressed as mean ± SEM. Statistical comparisons of all data were analysed by GraphPad Prism 6/7/8/9 (GraphPad Software, USA). *P* values were determined by unpaired two-tailed *t*-tests between two groups or three-way ANOVA, two-way ANOVA or one-way ANOVA followed by Tukey's or Dunnett's post hoc test between multiple groups, as indicated in the figure legends. The source of significant Three-way interactions was isolated with two-way ANOVA performed on each treatment group separately and Tukey post hoc tests. Sidak's multiple comparison post hoc test was used to assess significant differences in paw withdrawal thresholds at each time point in chronic dosing studies. $P < 0.05$ was considered statistically significant. No statistical method was used to predetermine the sample size. No data were excluded from the analyses. The investigators were blinded to pharmacological treatments in all behavioural studies. In all other experiments, investigators were not blinded.

### Reporting summary

Further information on research design is available in the Nature Portfolio Reporting Summary linked to this article.

## Data availability

Full-length cDNA encoding human MAGL (GenBank ID: BC006230.2), mouse and human LIPE (referred to as 'HSL' elsewhere in manuscript; GenBank IDs: BC021642 and BC070041 respectively) are available at GenBank. The co-crystal structure of MAGL and LEI-515 is available in the PDB using code 8AQF. The mass spectrometry proteomics data generated have been deposited to the ProteomeExchange Consortium via the PRIDE partner repository with the dataset identifier PXD042220. All other data needed to evaluate the conclusions in the paper are present in the paper, the Supplementary Materials and/or the source data file. Source data are provided with this paper.

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

## Acknowledgements

We thank Alex Makriyannis for gifting CB[1] antagonist AM6545. Funding for this research was provided by the Intramural Programme of NIAAA/NIH (P.P.), the Indiana Addiction Grand Challenges grants, DA047858, DA009158 (A.G.H.) and T32 DA042584 (J.W., A.G.H.), Netherlands Organization for Scientific Research VICI-grant, 724.017.002 & Onco-Drugs (M.J., M.S.) & Oncode Institute (M.S.). Research Travel Grant from the Karlsruhe House of Young Scientists + Graduate Funding from the German States (F.M.).

## Author contributions

Conceptualization: M.S., A.G.H., C.A.A.B., R.J.B.H.N.B., A.P.A.J., P.P. High throughput screening: T.W., T.W.L., H.H. Hit optimization: M.J., M.C.W.H., F.M., A.A. Crystallization: J.B. Chemical proteomics: M.J., B.I.F. Lipidomics: M.J., X.D., A.F.S., T.H. In situ activity and reversibility assay with LEI-463-Cy5: H.D., D.V. NAPE-PLD activity: W.F.D. CB1/2 activity: L.H.H. In vivo target engagement: A.P., I.R., M.B.W., L.C., U.G. Acute liver injury model: J.P., P.P. In vivo study: J.L.W., A.G.H. Writing—original draft: M.S., A.G.H., M.C.W.H., P.P., M.J.

## Competing interests

M.S., M.J., F.M., C.A.A.B., M.C.W.H., A.A. are inventors on a patent application related to this work filed by the University of Leiden (no. PCT/EP2021/055315, filed 3 March 2021). J.B., A.P., I.R., M.B.W., L.C. and U.W. are affiliated with, and M.S. is a consultant for, Hoffmann-La Roche Ltd. A.G.H. is a consultant for Anagin, Inc. The remaining authors declare no competing interests.
