## [Peer Review File · Nature Communications]

REVIEWER COMMENTS

Reviewer #1 (Remarks to the Author):

Potential of endocannabinoid signaling activity via inhibition of the serine hydrolase monoacylglycerol lipase (MAGL) is an appealing strategy in the development of treatments for several disorders, including ones related to mood, pain, and inflammation. The manuscript describes a peripherally restricted, reversible MAGL inhibitor (LEI-515) that attenuates inflammation and neuropathic nociception without side effects associated with direct CB1R activation. The importance of this type of compounds is more related with its use as working tool (to discriminate between central and peripheral functions of MAGL) rather than a clinical need, since it is widely accepted that MAGL inhibition with reversible inhibitors do not produce psychoactive effects, even if they reach the CNS. This notion is thoroughly supported by recent work, and some recent examples are included in J. Med. Chem. 2021, 64, 11014; J. Med. Chem. 2021, 64, 14283; J. Pharmacol. Exp. Ther. 2020, 372, 339; J. Med. Chem. 2020, 63, 5783.....). Also, several potent, reversible and selective MAGL inhibitors have been reported (J. Med. Chem. 2022, 65, 7118; J. Med. Chem. 2019, 62, 1932....). In addition, there are several important weaknesses that preclude publication in Nature Communications:

- Regarding the medicinal chemistry, the authors indicate that they have made a ligand-based drug design approach to improve the potency of compound 1. The chemical modifications from compound 1 to 3 are not guided by this design, they are classical modifications in a medicinal chemistry program.
- A summary of a structure-activity relationship study is missing.
- Although the synthesis of compounds 1-3 is detailed in the supporting information, it is not mentioned throughout the manuscript.
- A very important aspect that makes this work incomplete is that derivative 3 (LEI-515) is a racemic compound that is biologically evaluated. However, both enantiomers must be synthesized and evaluated pharmacologically, especially taking into account that in the crystallization study with the enzyme only one isomer crystallizes, the 2S,3S isomer.
- Levels of 2-AG, AA and AEA in heart, lung and kidney after oral administration of LEI-515 should be determined
- In relation with the previous point, to confirm peripheral distribution, and considering the in vivo models analyzed, it is important to determine: (i) the time course of the levels of LEI-515 in peripheral organs (at least in kidney, liver and lungs as representative peripheral tissues where the action of a MAGL inhibitor can be clinically useful) after the administration of the compound; and (ii) whether MAGL enzyme is actually inhibited in the above-mentioned peripheral tissues after administration of LEI-515.
- In the introduction the literature is heavily biased towards irreversible inhibitors. Citation of relevant previous work carried out regarding the development of potent, selective and reversible MAGL inhibitors with in vivo efficacy should be included in the introduction.

Reviewer #2 (Remarks to the Author):

The manuscript titled "Discovery of a monoacylglycerol lipase inhibitor that harnesses the therapeutic potential of endocannabinoid signaling without CNS-mediated adverse side effects or physical dependence", by Jiang et al., provide an exhaustive pharmacological characterization of a new discovered compound (called LEI-515). Using in vitro and in vivo assays, the Authors show the promising therapeutic potential of the LEI-515 compound in animal models of acute inflammation and chronic pain. Remarkably, the LEI-515 compound discovered by Authors, in contrast to others a monoacylglycerol lipase inhibitors, lacks in CNS-mediated adverse side effects or physical dependence.

Overall, the manuscript deals with a very interesting and innovative topic, but the results could be ameliorated in order to support the conclusions. Moreover, the Materials and Methods section should be improved.

Somme issues need to be addressed before publication in Nat comm journal.

The results showed in the chapter: LEI-515 is a peripherally restricted, orally bioavailable MAGL inhibitor, are not clear. The Authors should explain why they used the dosage of 10 mg/Kg in vivo.

The Authors should clarify why peripheral and brain target modulation was performed only for the LEI-515 oral administration. In addition, about the caption of the Figure 4 (related to the chapter), the Authors used: Unpaired t-test statistical analysis. Do the Authors also compared the groups with other statistical analysis. Moreover, the number of animals for vehicle does not correspond to the number of dots in the graph. Please, check and fix accordingly.

In chapter: LEI-515 displays efficacy in acute liver injury model (line 222), the Authors should insert the citation of the study about the use of JZL-184. In line 224, the Authors used a dosage of 30 mg/Kg without specify the route of administration. Please add the via of LEI-515 administration. Moreover, the Figure number related to the chapter, is incorrect. The results described in this chapter seems to fix with Figure.5. Please, check it. Line 988 is confusing, it could be rephrased. The micrograph of histological studies contains the control group (animal without CCI4) that has not been analyzed, please exclude it from the micrograph or report it also in the statistical analysis.

Considering the chapter LEI-515 displays cannabinoid CB2 receptor-dependent antinociceptive efficacy in a chemotherapy-induced neuropathic pain model, change allodynia and use mechanical hypersensitivity consistently in the line 238 and throughout the entire manuscript. The Authors should also add in the background the rationale about the use of both sexes: female and male mice. Line 257 that received the cremaphor vehicle in lieu of paclitaxel, typing error.

273 Typing error: but not in mice treated with LEI-515.

Line 675. The Authors should report the number of subjects used to perform the acute liver injury model. Considering the different treatments used, the Authors should specify that mice were group-housed within a cage received the same treatment. Moreover, the Authors should mention also the control group (animals without pathology).

683 (at 10 and 30 mg/kg doses) or its vehicle were administered intraperitoneally. Please, explain why only the ip administration was used and why were used these dosages of LEI-515.

702 In vivo assessment of LEI-515. Please add the number of mice used.

Caption Figure 1. Font typing error. Put in bold the letters in brackets (line 953 and 955). The Authors mentioned :Supplementary Fig. 1. and Supplementary Table 1 and 2. Perhaps I have missed it somewhere but I cannot readily find information on.

Caption Figure 3. Please add gel description also for graphic 3b and 3e

Reviewer #3 (Remarks to the Author):

Jiang et al. report a breakthrough in the search for a selective, reversible inhibitor for MAGL that is peripherally restricted and acts outside the CNS. In the current manuscript, they present an impressive amount of data starting from the search for the lead, to in vitro, in-cell and in vivo studies in mouse models supporting their claim.

First, they use HTS-screening to identify a very promising lead compound selected from a qualified hit and selective for MAGL over FAAH list with different chemotypes; This is followed by medicinal chemistry program (which will be described elsewhere) to improve potency and reduce potential toxicity of the nitrophenyl group in the original compound – which lead to LEI-515 as potent, metabolically stable MAGL inhibitor ($pIC_{50} = 9.3 \pm 0.1$). LEI-515 is covalently binding inhibitor as determined by a complex-crystal structure of a solubility enhanced variant of human MAGL with LEI-515. The activated ketone of the inhibitor LEI-515 binds covalently to the catalytic serine 122 and forms a hemiketal. Experiments are performed to test for the reversible mode-of-action and the effect of the inhibitor in living cells was then shown with respect to increase in 2-AG levels. The absorption, distribution, metabolism, and excretion profile of LEI-515 was determined and demonstrated a peripherally restricted MAGL inhibitor. Furthermore they showed that LEI-515 reduces negative effects in acute liver injury models similar to that observed with the toxic inhibitor JZL-184. Mice studies showed reduced chemotherapy-induced neuropathic nociception without CNS-mediated adverse effects.

The study is very comprehensive and impressive, and I considered it as highly interesting for the readers of nature communications.

Some aspects should be improved in the text before the recommending final acceptance of the ms. These include some discussion points and in part more in-depth description of the performed experiments.

Detailed comments/questions:

1) Figure 1: How much does LEI-515 affect activity of ABHD6 and ABHD12. Does FP-TAMRA also affect ABHD12? Please comment!

2) Preparation of Membrane fractions need to be described better; Transfection is well described, assay is very well described, but the critical preparation of protein preparation is lacking. What is the purity of the membrane fraction that were used? What concentration is estimated for MAGL in the assays – 0.3 ug per 100 ul?

3) Table S1: Results-HTS and hit triaging.

The authors state: “Subsequently, structures were examined and compounds with an apparent irreversible mode of action were excluded at this stage, as well as those with poor physicochemical properties or protection by intellectual property.” It is not clear how the irreversible mode of action was determined with the performed experiments. Please elaborate in more detail.

4) Crystallization studies: human MAGL Lys36Ala, Leu169Ser and Leu176Ser23 .. cloning, expression (cell type... E.coli. HEK cells, Sf9 cells?); TEV-cleavage? purification needs to be described. Is this Schalk-Hihi Protein Sci 2011? doi: 10.1002/pro.596

5) Deposition in PDB: 3PE6 ? the noted ‘PDB entry 3pe623’ is not a valid pdb-code? The crystal-structure determination part appears to be copied out from somewhere with previous references not left in 2-digit format (3PE623, PHASER33, COOT35 etc).

6) For the soaking experiment of LEI-515: Was LEI-515 dissolved in DMSO before adding it to the crystallization condition or was it added as powder, or is it liquid by itself? The covalent binding was still present after 16hrs of soaking. Did the researchers see a decrease in inhibitor in the structure? Are there any complex-structures available, where the bond-length is broken again? Please comment this with respect to reversible inhibition seen in mouse and human brain lysates.

7) Suppl Tabl3, vs text. slight mismatch in beta-angle for the space-group (90 or 90.57 in text); please also report CC1/2, units of B-factors

8) Figure 3_ LEI-515 is supposed to be a reversible inhibitor in peripheral organs outside the CNS, why are comparative studies of inhibitors (e.g. ABX-1431) only done in brain? Please comment?

9) Fig 3a. Please provide an insert mouse vs human as written in the text.

10) Fig. 3b: Include the use of both, LEI-463-Cy and LEI-515 as explained in the text to increase readability? Is the DMSO-control actually LEI-463-Cy in DMSO? Please also comment on the double band for MAGL in the chase experiment.

11) Figure 3c: Experiment in right panel was performed with MB064. Describe in the ms, which probe they can detect what? FAAH, ABHD12, MGL and ABHD6 are indicated on the right hand panel, but the heights are not consistent in the two blots.. please correct and comment; eg. Do FAAH and MAGL appear only with TAMRA-FP

12) Fig3d: give more details in the text and figure legend.

13) Figure 3e-3h: Which cells were used in these assays HS578t, U87-MG? please clarify.

- 14) 3g: Is the drop in 2-AG increase “real” for 0.5 and 1 μ M? Please comment briefly.
- 15) 3h: If plotted on two different panels the drastic reduction on AA compared to AEA would be much more obvious at a first glance. AEA needs to be introduced appropriately, ms says anandamide only.
- 16) Figure S3 left panel: please indicate the expected height for human HSL.
- 17) Line 199: “LEI-515 exhibited high human and mouse protein binding (99.6% for both).”

For better understanding it would be good to connect it to the plasma protein binding assay.

- 18) Figure 4: 2-AG, AA, AEA levels were measured upon LEI-515 treatment. The limited role of HSL in 2-AG production is discussed. How about MG-levels more likely derived from lipid metabolism? HSL being also partly inhibited, is every MG-hydrolytic activity compensated by ABHD6/12 in lung and liver? Can you bring Fig S5a as extra panel to the main manuscript?
- 19) From Line 220/Figure 5: Chapter - LEI-515 displays efficacy in acute liver injury model. Figure numbers do not match! Please correct!
- 20) Figure 6: MPE on the Y axis. Please explain in more detail in the text.
- 21) Supplemental data should include expression details for human and mouse HSL, which have been notoriously difficult enzymes to produce. Methods briefly mention LIPE, please indicate that this refers to cds coding for HSL in the text. Human and mouse LIPE isoforms typically differ by approx 300 amino acids. Please give reference (GeneBank, Uniprot) what genes were synthesized. Ref 23 is not a good source to find this information.
- 22) Figure S4: 13 breast cancer cell lines were investigated, MGL protein levels are quite different. Has this been shown before, e.g. different chromosomal copy numbers? Why do some ABPP data look some much stronger compared to their expression profile (e.g. SUM229PE?) Why is the lower band predominantly picked up by ABPP. Please comment. LEI-463 in figure legend refers to LEI-463-Cy5? What antibody was used for WB?

TYPOS/SMALL CHANGES

Some Figure labels/descriptions are difficult to read due to small font size (e.g. top of Fig3e) please improve those throughout the ms;

Figure captions in text and figures should be uniform (A) or a) .. eg Figure 4,

Spelling is not consistent: JZL184 (JZL-184), WIN55,212-2 (WIN 55,212-2), Lei515 (LEI-515), 4xLaemmli buffer (4*Laemmli-buffer).

For clarity to the non-medicinal chemistry audience, the definition of pIC50 should be stated (at least at first time usage), since inhibitory constants are associated with concentrations, %MPE.

Reviewer #4 (Remarks to the Author):

There are several issues:

In the figure 3b, the full recovery of MAGL activity with LEI515 inhibitor is not visible in the time curve shown. It should be shown MAGL activity at time 0 with LEI-515 and after 120 minutes to see the full recovery of MAGL activity.

In line 173, authors should explain deeply Figure 3c and also in figure legend.

In figure 3 f, Authors analyze the time-dependent incubation with LEI-515, and in figure 3g, they also analyze the concentration-dependent incubation at 1h. Why have they selected 1h to study LEI515 doses? Have they tried other times?

Indeed, in figure 3h, authors observed a decrease in AA and AEA levels in cells treated with LEI-515 (1h, 1uM). Have they evaluated AA and AEA levels with other doses of LEI-515 and other incubation-times?

In line 214, authors affirm that LEI-515 induced a concentration-dependent increase in 2-AG in the intestine, lung and liver, but not in brain. However, there is no significant difference in the liver (Supp Fig 5), so it is not possible to affirm that sentence.

In line 225, authors incorrectly reference Fig 4c-e.

In lines 220-231, authors should check this paragraph. Figures are incorrectly referenced. Fig 4e,f g and h do not exist and they do not mention figure 5 in the text.

Minor revisions:

In line 109, authors should include the corresponding reference of the study in which benzoxazine derivates and piperazine amides were reported as MAGL inhibitors.

In line 124, authors should reference table S2 in which the different compounds are described. Moreover, I cannot find the reference of table S2 in the manuscript. I suggest to describe the name of each compound in table S2.

I strongly recommend to restructure the position of the panels in figure 1 and especially figure 3.

Authors should revise the outliers and repeat the statistical analysis in some figures: Supp Fig 5d-e, Figure 4d, Figure 5a and Figure 6a.

Authors should specify coomassie bands in Supp. Fig 3

Authors should check the number of replicates or samples used in each condition for the different experiments. The number should be similar. For example, in figure 5, 8 mice were used for the acute liver injury model, but in fig 5a the number of animales is higher than 8.

REVIEWER COMMENTS

Reviewer #1 (Remarks to the Author):

Potential of endocannabinoid signaling activity via inhibition of the serine hydrolase monoacylglycerol lipase (MAGL) is an appealing strategy in the development of treatments for several disorders, including ones related to mood, pain, and inflammation. The manuscript describes a peripherally restricted, reversible MAGL inhibitor (LEI-515) that attenuates inflammation and neuropathic nociception without side effects associated with direct CB1R activation. The importance of this type of compounds is more related with its use as working tool (to discriminate between central and peripheral functions of MAGL) rather than a clinical need, since it is widely accepted that MAGL inhibition with reversible inhibitors do not produce psychoactive effects, even if they reach the CNS. This notion is thoroughly supported by recent work, and some recent examples are included in J. Med. Chem. 2021, 64, 11014; J. Med. Chem. 2021, 64, 14283; J. Pharmacol. Exp. Ther. 2020, 372, 339; J. Med. Chem. 2020, 63, 5783.....). Also, several potent, reversible and selective MAGL inhibitors have been reported (J. Med. Chem. 2022, 65, 7118; J. Med. Chem. 2019, 62, 1932.....).

Response to comment 1: We respectfully disagree with the reviewer. There is no scientific explanation why CNS-permeable, reversible MAGL inhibitors would not induce psychoactive effects. Those compounds will elevate 2-AG levels in the brain and modulate neurotransmission via CB1R activation. Depending on the dose and the number of administrations this may lead to unwanted CNS adverse effects. The compounds reported in the papers cited by the reviewer are all brain penetrant MAGL inhibitors that modulate CNS functions. We report here the first peripherally, restricted MAGL inhibitor with anti-nociceptive effects in a chemotherapy-induced neuropathy model that does not induce any CNS-adverse effects or physical dependence.

In addition, there are several important weaknesses that preclude publication in Nature Communications:

- Regarding the medicinal chemistry, the authors indicate that they have made a ligand-based drug design approach to improve the potency of compound 1. The chemical modifications from compound 1 to 3 are not guided by this design, they are classical modifications in a medicinal chemistry program.

Response to comment 2: We respectfully disagree with the reviewer. Ligand-based drug design is defined as a drug discovery approach to identify potent ligands for a protein based on the rational modification of the structure of a series of ligands in the absence of three dimensional structural information of the target protein (as opposed to structure-based drug design that uses a co-crystal structure, for example). [The Practice of Medicinal Chemistry, Wermuth et al., 4ed, 2015]

- A summary of a structure-activity relationship study is missing.

Response to comment 3: We have uploaded a separate manuscript (posted on ChemRxiv) detailing the structure-activity relationship of LEI-515, which is included in this resubmission.

- Although the synthesis of compounds 1-3 is detailed in the supporting information, it is not mentioned throughout the manuscript.

Response to comment 4: We have now included the synthesis and analytical characterization of compounds 1-3 in the SI.

- A very important aspect that makes this work incomplete is that derivative 3 (LEI-515) is a racemic compound that is biologically evaluated. However, both enantiomers must be synthesized and evaluated pharmacologically, especially taking into account that in the crystallization study with the enzyme only one isomer crystallizes, the 2S,3S isomer

Response to comment 5: We respectfully disagree with the reviewer. We have indicated that the compound series, including LEI-515, is racemic (Figure 2a and SI). In line with the author guidelines of Nature Comm., we have provided full analytical characterization (¹H-NMR, ¹³C-NMR and HRMS) of

the compounds (in the SI). The synthesis and biological characterization of the separate enantiomers is outside the scope of the current manuscript.

- Levels of 2-AG, AA and AEA in heart, lung and kidney after oral administration of LEI-515 should be determined

*Response to **comment 6**: We have reported the levels of 2-AG in four different tissues (including lung, colon, liver and brain; Figure 4B and S5). To study target modulation for a covalent, reversible inhibitor such as LEI-515 in mouse it is common strategy to study whether the ligand reaches a set of representative, relevant and PK-wise differently behaving organs and exerts its action (i.e. via measurement of 2-AG levels), which we have successfully demonstrated. Since LEI-515 distributes well in the periphery ($V_{ss} = 2.1$ L/kg), adding heart and kidney, which are well-perfused organs with no specific barriers for drugs, will not provide any new information. Arguably more importantly, these organs are not the key drivers for the observed pharmacodynamic effects in the neuropathy and cirrhosis models, thus there is no additional scientific value in measuring 2-AG levels in kidney and heart (Cao et al., Gastroenterology, 2013, 144, 808; Lin et al., Pain, 2022, 163, 834). Consequently, we consider these proposed extra animal experiments as non-essential and non-ethical for the current manuscript, because they will not change the conclusions.*

- In relation with the previous point, to confirm peripheral distribution, and considering the in vivo models analyzed, it is important to determine: (i) the time course of the levels of LEI-515 in peripheral organs (at least in kidney, liver and lungs as representative peripheral tissues where the action of a MAGL inhibitor can be clinically useful) after the administration of the compound; and (ii) whether MAGL enzyme is actually inhibited in the above-mentioned peripheral tissues after administration of LEI-515.

*Response to **comment 7**: We have reported a full PK curve over a 24h period for LEI-515 (Figure 4a and Table S5). The compound reached its T_{max} at 0,5 h with a C_{max} of 2433 nM after oral dosing. After 24h, LEI-515 has a plasma concentration of 100 nM, which is 200x higher than its IC_{50} . Combined with the target modulation data (2-AG levels) in the intestines, liver, lung and brain at 1h and the efficacy data in our neuropathy model at 0.5, 2, 6 and 24 h; we have demonstrated that LEI-515 is acting as a peripheral restricted MAGL inhibitor. Additional time points in kidney, liver and lung will not add any new information that is relevant nor necessary to interpret and explain the effects of LEI-515 in the neuropathy model. Consequently, we view these proposed extra animal experiments as non-essential and non-ethical for the current manuscript, because they will not change the conclusions.*

- In the introduction the literature is heavily biased towards irreversible inhibitors. Citation of relevant previous work carried out regarding the development of potent, selective and reversible MAGL inhibitors with in vivo efficacy should be included in the introduction.

*Response to **comment 8**: We have added the following references:*

22. Aida, J. et al. Design, Synthesis, and Evaluation of Piperazinyl Pyrrolidin-2-ones as a Novel Series of Reversible Monoacylglycerol Lipase Inhibitors. *J. Med. Chem.* **61**, 9205–9217 (2018).
23. Granchi, C. et al. Optimization of a Benzoylpiperidine Class Identifies a Highly Potent and Selective Reversible Monoacylglycerol Lipase (MAGL) Inhibitor. *J. Med. Chem.* **62**, 1932–1958 (2019).
24. Wyatt, R. M. et al. Pharmacologic characterization of JNJ-42226314, [1-(4-Fluorophenyl)indol-5-yl]-[3-[4-(thiazole-2-carbonyl) piperazin-1-yl]azetid-1-yl]methanone, a reversible, selective, and potent monoacylglycerol lipase inhibitor. *J. Pharmacol. Exp. Ther.* **372**, 339–353 (2020).
25. Zhi, Z. et al. Discovery of Aryl Formyl Piperidine Derivatives as Potent, Reversible, and Selective Monoacylglycerol Lipase Inhibitors. *J. Med. Chem.* **63**, 5783–5796 (2020).
26. Bononi, G. et al. Reversible Monoacylglycerol Lipase Inhibitors: Discovery of a New Class of

Benzylpiperidine Derivatives. *J. Med. Chem.* (2022). doi:10.1021/acs.jmedchem.1c01806

27. Granchi, C. *et al.* Design, synthesis and biological evaluation of second-generation benzoylpiperidine derivatives as reversible monoacylglycerol lipase (MAGL) inhibitors. *Eur. J. Med. Chem.* **209**, 112857 (2021).
28. Ikeda, S. *et al.* Design and Synthesis of Novel Spiro Derivatives as Potent and Reversible Monoacylglycerol Lipase (MAGL) Inhibitors: Bioisosteric Transformation from 3-Oxo-3,4-dihydro-2 H-benzo[b][1,4]oxazin-6-yl Moiety. *J. Med. Chem.* **64**, 11014–11044 (2021).

Reviewer #2 (Remarks to the Author):

The manuscript titled “Discovery of a monoacylglycerol lipase inhibitor that harnesses the therapeutic potential of endocannabinoid signaling without CNS-mediated adverse side effects or physical dependence”, by Jiang *et al.*, provide an exhaustive pharmacological characterization of a new discovered compound (called LEI-515). Using *in vitro* and *in vivo* assays, the Authors show the promising therapeutic potential of the LEI-515 compound in animal models of acute inflammation and chronic pain. Remarkably, the LEI-515 compound discovered by Authors, in contrast to others a monoacylglycerol lipase inhibitors, lacks in CNS-mediated adverse side effects or physical dependence. Overall, the manuscript deals with a very interesting and innovative topic, but the results could be ameliorated in order to support the conclusions. Moreover, the Materials and Methods section should be improved. Some issues need to be addressed before publication in *Nat comm journal*.

The results showed in the chapter: LEI-515 is a peripherally restricted, orally bioavailable MAGL inhibitor, are not clear. The Authors should explain why they used the dosage of 10 mg/Kg *in vivo*.

Response to comment 9: This is a standard dose to study the pharmacokinetic properties and efficacy of a compound after oral dosing. See for example our previous studies (Mock *et al.*, *Nature Chem Biol.*, 2020, 16, 667 and Li *et al.*, *Nature Comm.*, 2023, 14, 1447).

The Authors should clarify why peripheral and brain target modulation was performed only for the LEI-515 oral administration.

Response to comment 10: It is important to show a therapeutic window between CNS-side effects and efficacy via the preferred route of administration. The *i.v.*-administration was only used in our study to determine the primary pharmacokinetic parameters (Clearance and V_{ss}), the half life of the compound and the oral bioavailability. Therefore, we have only determined brain penetration after oral administration, because that is the preferred route of administration in the neuropathy model.

In addition, about the caption of the Figure 4 (related to the chapter), the Authors used: Unpaired t-test statistical analysis. Do the Authors also compared the groups with other statistical analyses. Moreover, the number of animals for vehicle does not correspond to the number of dots in the graph. Please, check and fix accordingly.

Response to comment 11: Thank you for noticing. We have checked and fixed the figure.

In chapter: LEI-515 displays efficacy in acute liver injury model (line 222), the Authors should insert the citation of the study about the use of JZL-184.

Response to comment 12: We have inserted Cao Z, Mulvihill MM, Mukhopadhyay P, Xu H, Erdélyi K, Hao E, Holovac E, Haskó G, Cravatt BF, Nomura DK, Pacher P. *Gastroenterology*. 2013;144(4):808-817.

In line 224, the Authors used a dosage of 30 mg/Kg without specify the route of administration. Please add the via of LEI-515 administration.

Response to comment 13: Thank you for noticing this omission. It was administered *i.p.*.

Moreover, the Figure number related to the chapter, is incorrect. The results described in this chapter seems to fix with Figure.5. Please, check it.

Response to comment 14: Thank you for noticing it. We have corrected the figure numbers.

Line 988 is confusing, it could be rephrased. The micrograph of histological studies contains the control group (animal without CCI4) that has not been analyzed, please exclude it from the micrograph or report it also in the statistical analysis.

Response to comment 15: In the revised figure we eliminated control groups. We also considerably increased the n numbers including both sexes. In addition, we also including a supplemental figure 6 showing that 30 mg/kg LEI515 i.p. has no effect on liver enzymes or liver pathology in normal mice.

Considering the chapter LEI-515 displays cannabinoid CB2 receptor-dependent antinociceptive efficacy in a chemotherapy-induced neuropathic pain model, change allodynia and use mechanical hypersensitivity consistently in the line 238 and throughout the entire manuscript.

Response to comment 16: We have adapted the text.

The Authors should also add in the background the rationale about the use of both sexes: female and male mice.

Response to comment 17: In line with previous studies from our lab and guidelines of Nature Communications, we have studied the antinociceptive effect in both sexes. Deng, L., Mol Pharmacol, 2015; Deng, L, Biol Psychiatry, 2015; Lin, X. Pain, 2022.

Line 257 that received the cremaphor vehicle in lieu of paclitaxel, typing error.
273 Typing error: but not in mice treated with LEI-515.

Response to comment 18: Thank you for noticing. We have corrected the typo.

Line 675. The Authors should report the number of subjects used to perform the acute liver injury model. Considering the different treatments used, the Authors should specify that mice were group-housed within a cage received the same treatment. Moreover, the Authors should mention also the control group (animals without pathology).

Response to comment 19: For the acute liver injury study 64 mice were used (39 males and 25 females). Mice group-housed within a cage (3-4 mice/cage) received the same treatment. 40 mice were used for liver injury study (CCL4) and 24 for controls. The effect of 30 mg/kg LEI515 i.p. in control mice on liver enzymes and pathology is shown in supplemental figure 6.

Line 683 (at 10 and 30 mg/kg doses) or its vehicle were administered intraperitoneally. Please, explain why only the ip administration was used and why were used these dosages of LEI-515.

Response to comment 20: Previously, the central penetrant, covalent irreversible MAGL inhibitor JZL184 was able to attenuate acute liver injury induced by CCl₄ (Cao et al. Gastroenterology 2013). Here, we wanted to assess whether our reversible MAGL inhibitor LEI-515 was able to replicate these findings. Therefore, we selected comparable doses of LEI-515 and the same route of administration to be able to compare LEI-515 with JZL184.

Line 702 In vivo assessment of LEI-515. Please add the number of mice used.

Response to comment 21: We have added the number of mice used.

Caption Figure 1. Font typing error. Put in bold the letters in brackets (line 953 and 955).

Response to comment 22: corrected.

The Authors mentioned :Supplementary Fig. 1. and Supplementary Table 1 and 2. Perhaps I have missed it somewhere but I cannot readily find information on.

*Response to **Comment 23**: The supplementary Fig. 1. and Table S1 and S2 can be found in the supporting information.*

Caption Figure 3. Please add gel description also for graphic 3b and 3e

*Response to **Comment 24**: We have changed the gel description in the figure and added more details in the figure legends.*

Reviewer #3 (Remarks to the Author):

Jiang et al. report a breakthrough in the search for a selective, reversible inhibitor for MAGL that is peripherally restricted and acts outside the CNS. In the current manuscript, they present an impressive amount of data starting from the search for the lead, to in vitro, in-cell and in vivo studies in mouse models supporting their claim.

First, they use HTS-screening to identify a very promising lead compound selected from a qualified hit and selective for MAGL over FAAH list with different chemotypes; This is followed by medicinal chemistry program (which will be described elsewhere) to improve potency and reduce potential toxicity of the nitrophenyl group in the original compound – which lead to LEI-515 as potent, metabolically stable MAGL inhibitor ($pIC_{50} = 9.3 \pm 0.1$). LEI-515 is covalently binding inhibitor as determined by a complex-crystal structure of a solubility enhanced variant of human MAGL with LEI-515. The activated ketone of the inhibitor LEI-515 binds covalently to the catalytic serine 122 and forms a hemiketal. Experiments are performed to test for the reversible mode-of-action and the effect of the inhibitor in living cells was then shown with respect to increase in 2-AG levels. The absorption, distribution, metabolism, and excretion profile of LEI-515 was determined and demonstrated a peripherally restricted MAGL inhibitor. Furthermore they showed that LEI-515 reduces negative effects in acute liver injury models similar to that observed with the toxic inhibitor JZL-184. Mice studies showed reduced chemotherapy-induced neuropathic nociception without CNS-mediated adverse effects. The study is very comprehensive and impressive, and I considered it as highly interesting for the readers of nature communications.

Some aspects should be improved in the text before the recommending final acceptance of the ms. These include some discussion points and in part more in-depth description of the performed experiments.

Detailed comments/questions:

1) Figure 1: How much does LEI-515 affect activity of ABHD6 and ABHD12. Does FP-TAMRA also affect ABHD12? Please comment!

*Response to **comment 25**: During hit optimization we regularly tested the selectivity over ABHD6 and ABHD12 using ABPP with MB064 as a broad-spectrum probe in mouse brain lysates (Figure 3c). LEI-515 is selective over ABHD6 and ABHD12 (Figure 3c), which was confirmed by the chemical proteomics experiments (Figure 3d). FP-TAMRA does not label ABHD12 very well. Hence, we have used a gel-based ABPP protocol to comprehensively visualize the enzymes of the endocannabinoid system to determine the selectivity of the inhibitors. See Janssen APA et al. Development of a Multiplexed Activity-Based Protein Profiling Assay to Evaluate Activity of Endocannabinoid Hydrolase Inhibitors. ACS Chem Biol. 2018;13(9):2406-2413.*

2) Preparation of Membrane fractions need to be described better; Transfection is well described, assay is very well described, but the critical preparation of protein preparation is lacking. What is the purity of the membrane fraction that were used? What concentration is estimated for MAGL in the assays – 0.3 ug per 100 ul?

*Response to **comment 26**: We added information on how the membrane fractions were prepared. We did not assess the purity of the membrane. To correct for background 2-AG hydrolysis we used overexpressed mock membranes that were prepared in the same manner as the MAGL overexpression membranes but instead an empty plasmid was used. The estimated concentration for MAGL in the assay 3 ng/μL during the incubation step, and 1.5 ng/μL after addition of the assay mix.*

3) Table S1: Results-HTS and hit triaging.

The authors state: "Subsequently, structures were examined and compounds with an apparent irreversible mode of action were excluded at this stage, as well as those with poor physicochemical properties or protection by intellectual property." It is not clear how the irreversible mode of action was determined with the performed experiments. Please elaborate in more detail.

*Response to **comment 27**: "..., judged by the presence of chemical motifs commonly found in irreversible serine hydrolase inhibitors, such as lactones or activated carbamates/ureas." We added this sentence in line 106.*

4) Crystallization studies: human MAGL Lys36Ala, Leu169Ser and Leu176Ser23 .. cloning, expression (cell type... E.coli. HEK cells, Sf9 cells?); TEV-cleavage? purification needs to be described. Is this Schalk-Hihi Protein Sci 2011? doi: 10.1002/pro.596

*Response to **comment 28**: This information is added in the experimental section of the crystallization studies (line 457). This is indeed according to Schalk-Hihi, Protein Sci, 2011 (citation 50).*

5) Deposition in PDB: 3PE6 ? the noted 'PDB entry 3pe623' is not a valid pdb-code? The crystal-structure determination part appears to be copied out from somewhere with previous references not left in 2-digit format (3PE623, PHASER33, COOT35 etc).

*Response to **comment 28**: Thank you for noticing, we have now corrected it.*

6) For the soaking experiment of LEI-515: Was LEI-515 dissolved in DMSO before adding it to the crystallization condition or was it added as powder, or is it liquid by itself? The covalent binding was still present after 16hrs of soaking. Did the researchers see a decrease in inhibitor in the structure? Are there any complex-structures available, where the bond-length is broken again? Please comment this with respect to reversible inhibition seen in mouse and human brain lysates.

*Response to **comment 29**: We collected data for a number of crystals but only at one time point with 16h soaking time and with high ligand concentrations. In all collected datasets the ligand occupancy is high and the electron density confirms the presence of the covalent bond under the conditions used in crystallization. We don't have structures determined in a time course experiment, with reduced ligand soaking concentrations or in experiments we tried to wash away unbound inhibitor from the crystal, because our goal was to characterize the molecular interactions of the ligand with the protein and not the reversible nature of LEI-515 in crystals.*

7) Suppl Tabl3, vs text. slight mismatch in beta-angle for the space-group (90 or 90.57 in text); please also report CC1/2, units of B-factors

*Response to **comment 30**: Corrected*

8) Figure 3_ LEI-515 is supposed to be a reversible inhibitor in peripheral organs outside the CNS, why are comparative studies of inhibitors (e.g. ABX-1431) only done in brain? Please comment?

*Response to **comment 31**: The *in vitro* characterization (selectivity / reversibility) studies of LEI-515 were performed, before we determined the mouse PK and brain permeability of the compound (in order to select the best compound). Thus, at that time we compared the effects of LEI-515 with ABX-1431 in mouse brain lysates, because other studies reported in literature study serine hydrolase selectivity in mouse brain proteome (e.g. Van Rooden et al., Nature Protocol., 2018). When we*

realized that LEI-515 was a peripherally restricted inhibitor (after the mouse PK study), we performed additional selectivity studies using chemical proteomics in lung (Figure 3d) and liver (Figure S2) proteomes.

9) Fig 3a. Please provide an insert mouse vs human as written in the text.

Response to comment 32: Corrected

10) Fig. 3b: Include the use of both, LEI-463-Cy and LEI-515 as explained in the text to increase readability? Is the DMSO-control actually LEI-463-Cy in DMSO? Please also comment on the double band for MAGL in the chase experiment.

Response to comment 33: We changed the figure and description: 'Time-dependent recovery of MAGL activity in mouse brain proteome of LEI-515 and ABX-1431 as determined by competitive ABPP using a MAGL selective probe LEI-463-Cy5 (1 μ M). Data represented as the fold change in MAGL labeling of the two isoforms as compared to the five minute chase time (n=3). ' The DMSO control is DMSO + LEI-463-Cy5 in DMSO, because both LEI-515 and ABX-1431 are also dissolved in DMSO. This way every sample had the same amount of DMSO. MAGL is known to exist in two splicing isoforms in specific tissues (Karlsson M, et al., Exon-intron organization and chromosomal localization of the mouse monoglyceride lipase gene. Gene. 2001;272:11–18)

11) Figure 3c: Experiment in right panel was performed with MB064. Describe in the ms, which probe they can detect what? FAAH, ABHD12, MGL and ABHD6 are indicated on the right hand panel, but the heights are not consistent in the two blots.. please correct and comment; eg. Do FAAH and MAGL appear only with TAMRA-FP

Response to comment 34: We have changed the figure according to the suggestions. FAAH and MAGL are detected only by FP-TAMRA, whereas ABHD6 and ABHD12 are readily visualized by MB064. We have previously developed a gel-based ABPP protocol using a probe cocktail to comprehensively visualize most enzymes of the endocannabinoid system, which we apply routinely in our studies. See Janssen APA et al. Development of a Multiplexed Activity-Based Protein Profiling Assay to Evaluate Activity of Endocannabinoid Hydrolase Inhibitors. ACS Chem Biol. 2018;13(9):2406-2413.

12) Fig3d: give more details in the text and figure legend.

Response to comment 35: We changed figure description to give more details on the probes including the concentrations.

13) Figure 3e-3h: Which cells were used in these assays HS578t, U87-MG? please clarify.

Response to comment 36: In figure 3e U87-MG cells were used. In Figure 3f – h HS578t cells were used.

14) 3g: Is the drop in 2-AG increase “real” for 0.5 and 1 μ M? Please comment briefly.

Response to comment 37: Drop in 2-AG increase is mostly likely not a “real” biological effect, but due to experimental variability as judged by the error bars.

15) 3h: If plotted on two different panels the drastic reduction on AA compared to AEA would be much more obvious at a first glance. AEA needs to be introduced appropriately, ms says anandamide only.

Response to comment 38: Thank you for the suggestion. AA plot is now Figure 3h and AEA is Figure 3i. Also AEA is appropriately introduced.

16) Figure S3 left panel: please indicate the expected height for human HSL.

Response to comment 39: done

17) Line 199: "LEI-515 exhibited high human and mouse protein binding (99.6% for both)." For better understanding it would be good to connect it to the plasma protein binding assay.

Response to comment 40: done

18) Figure 4: 2-AG, AA, AEA levels were measured upon LEI-515 treatment. The limited role of HSL in 2-AG production is discussed. How about MG-levels more likely derived from lipid metabolism? HSL being also partly inhibited, is every MG-hydrolytic activity compensated by ABHD6/12 in lung and liver? Can you bring Fig S5a as extra panel to the main manuscript?

Response to comment 41: Thank you for the interesting suggestion. We have focused in our studies on 2-AG as an endogenous ligand of the cannabinoid CB₁ and CB₂ receptors, because other monoglycerides (MG), derived from lipid metabolism, do not activate the CB receptors. MAGL is indeed able to hydrolyze also other MGs, but HSL has a much broader substrate scope, including TAGs, DAGs and MGs. HSL prefers shorter acyl chains over longer ones, where length is more important than saturation (Fredrikson, G., et al, J. Biol. Chem., 1981, 132, 1; Rodriques, J.A., et al, Biochim Biophys Acta Mol Cell Biol Lipids BBA-mol cell biol I, 2010, 1801, 1). Long MGs (> C₁₂) are also substrates for ABHD6 and ABHD12, which may compensate for MAGL/HSL inhibition in the liver and lung. Shorter MGs (< C₁₂) can be hydrolyzed by carboxylic esterases, which may also compensate for MAGL/HSL inhibition. However, since we did not measure other MGs, it is difficult to draw any conclusions on the effect of MAGL/HSL inhibition on the other MGs in the various tissues. We prefer to have Figure S5 in the SI, because we have already many figure panels in the main manuscript.

19) From Line 220/Figure 5: Chapter - LEI-515 displays efficacy in acute liver injury model. Figure numbers do not match! Please correct!

Response to comment 42: Thank you for noticing, we have corrected the figure numbers.

20) Figure 6: MPE on the Y axis. Please explain in more detail in the text.

Response to comment 43: We added an explanation in the figure legend.

21) Supplemental data should include expression details for human and mouse HSL, which have been notoriously difficult enzymes to produce. Methods briefly mention LIPE, please indicate that this refers to cds coding for HSL in the text. Human and mouse LIPE isoforms typically differ by approx 300 amino acids. Please give reference (GeneBank, Uniprot) what genes were synthesized. Ref 23 is not a good source to find this information.

Response to comment 44: The requested information was added to the materials & methods section (line 390)

22) Figure S4: 13 breast cancer cell lines were investigated, MGL protein levels are quite different. Has this been shown before, e.g. different chromosomal copy numbers? Why do some ABPP data look some much stronger compared to their expression profile (e.g. SUM229PE?) Why is the lower band predominantly picked up by ABPP. Please comment. LEI-463 in figure legend refers to LEI-463-Cy5? What antibody was used for WB?

*Response to **comment 45**: It has been reported that MAGL protein levels are different in hepatocellular carcinoma cell lines (Zhu, W, et al, J Hematol Oncol., 2016). Data showed that the protein and mRNA expression of MAGL in hepatocellular carcinoma cell lines (HepG2, SMMC7721, Huh7, MHCC97L, MHCC97H, and HCCLM3) were different compared to each other and in normal liver cell line (L0-2). Besides, in 2010 Nomura et al revealed that MAGL activity profile are also different in aggressive and nonaggressive cancer cells. The two forms of MAGL were detected in ABPP, are most likely alternative splicing forms of MAGL (Karlsson, M. et al, Gene, 2001). A potential explanation could be that one of the isoforms might be more sensitive towards oxidation that leads to an inactive state and cannot be detected by activity-based probes (T. Laitinen, et al, Mol Pharmacol., 2014, 83, 3; S. Tyukhtenko, et al, J Biol Chem., 2015, 85, 5; L. Scavini, et al, Chem Phys Lipids, 2016, 197, 13-24).*

Indeed, LEI-463 in figure S4 refers to LEI-463-Cy5. Thank you for noticing, we have changed it accordingly. Moreover, the antibody we used was ab24701. This information is also added to the figure legend.

TYPOS/SMALL CHANGES

Some Figure labels/descriptions are difficult to read due to small font size (e.g. top of Fig3e) please improve those throughout the ms;

Figure captions in text and figures should be uniform (A) or a) .. eg Figure 4,

Spelling is not consistent: JZL184 (JZL-184), WIN55,212-2 (WIN 55,212-2), Lei515 (LEI-515), 4xLaemmli buffer (4*Laemmli-buffer).

For clarity to the non-medicinal chemistry audience, the definition of pIC₅₀ should be stated (at least at first time usage), since inhibitory constants are associated with concentrations, %MPE.

*Response to **comment 46**: Thank you for the suggestions. We have corrected the typos. The definition of pIC₅₀ is stated in line 115.*

Reviewer #4 (Remarks to the Author):

There are several issues:

In the figure 3b, the full recovery of MAGL activity with LEI515 inhibitor is not visible in the time curve shown. It should be shown MAGL activity at time 0 with LEI-515 and after 120 minutes to see the full recovery of MAGL activity.

*Response to **comment 47**: After 120 min no full recovery of MAGL activity will be regained, because there is still inhibitor present in the incubation mixture. However, a clear time dependent increase in activity is observed whereas this is not the case for ABX-1431. Indicating a reversible binding mode of LEI-515.*

In line 173, authors should explain deeply Figure 3c and also in figure legend.

*Response to **comment 48**: We added the explanation in line 173, including the type of experiment and what probes were used. Moreover, the figure legend contains more information now on what probes were used at what concentrations and what the gel description means. For additional information we would like to refer to our paper by Janssen APA et al. Development of a Multiplexed Activity-Based Protein Profiling Assay to Evaluate Activity of Endocannabinoid Hydrolase Inhibitors. ACS Chem Biol. 2018;13(9):2406-2413, where these experimental protocols have been developed.*

In figure 3 f, Authors analyze the time-dependent incubation with LEI-515, and in figure 3g, they also analyze the concentration-dependent incubation at 1h. Why have they selected 1h to study LEI515 doses? Have they tried other times?

Indeed, in figure 3h, authors observed a decrease in AA and AEA levels in cells treated with LEI-515

(1h, 1uM). Have they evaluated AA and AEA levels with other doses of LEI-515 and other incubation-times?

*Response to **comment 49**: In Figure 1f, where we studied the time-dependency of MAGL inhibition, maximum effect was reached after 1h, therefore this time point was chosen for the concentration-dependent study (Figure 1g). We have not tested other time points and only evaluated the AA & AEA levels at 1h and 1 uM (Figure 1h and 1i), which resulted in significant biological effects that demonstrated that LEI-515 was cellular active.*

In line 214, authors affirm that LEI-515 induced a concentration-dependent increase in 2-AG in the intestine, lung and liver, but not in brain. However, there is no significant difference in the liver (Supp Fig 5), so it is not possible to affirm that sentence.

*Response to **comment 50**: Thank you for noticing. There is a significant correlation with 2-AG increase and LEI-515 concentration, albeit not with dose, in the liver. This could be due to PK properties of the compound (solubility and membrane permeability) or the expression of other 2-AG metabolizing enzymes (e.g. ABHD6, ABHD12, CESS). We should have been more precisely with our wording. We have adapted the statement accordingly.*

In line 225, authors incorrectly reference Fig 4c-e.

*Response to **comment 51**: Thank you for noticing. We have corrected the referencing.*

In lines 220-231, authors should check this paragraph. Figures are incorrectly referenced. Fig 4e,f g and h do not exist and they do not mention figure 5 in the text.

*Response to **comment 52**: Thank you for noticing. We have corrected the referencing.*

Minor revisions:

In line 109, authors should include the corresponding reference of the study in which benzoxazine derivates and piperazine amides were reported as MAGL inhibitors.

*Response to **comment 53**: We have added the references.*

In line 124, authors should reference table S2 in which the different compounds are described. Moreover, I cannot find the reference of table S2 in the manuscript. I suggest to describe the name of each compound in table S2.

*Response to **comment 54**: We have added the reference of table S2. The first column describes the cluster and the structure is given.*

I strongly recommend to restructure the position of the panels in figure 1 and especially figure 3.

*Response to **comment 55**: We have restructured the position of the panels in figure 3.*

Authors should revise the outliers and repeat the statistical analysis in some figures: Supp Fig 5d-e, Figure 4d, Figure 5a and Figure 6a.

*Response to **comment 56**: Additional animal experiments (CCl₄-study) were performed also to include female mice and the statistics was double-checked. The detailed statistical analysis also included in the raw data (Supplemental excel table)*

Authors should specify coomassie bands in Supp. Fig 3

*Response to **comment 57**: Corrected*

Authors should check the number of replicates or samples used in each condition for the different experiments. The number should be similar. For example, in figure 5, 8 mice were used for the acute liver injury model, but in fig 5a the number of animals is higher than 8.

*Response to **comment 58**: Thank you for noticing. The numbers were double-checked and corrected.*

REVIEWERS' COMMENTS

Reviewer #1 (Remarks to the Author):

In this revised version the authors have improved the initial work and hence I consider it is now suitable for publication in Nature Communications.

Reviewer #2 (Remarks to the Author):

The Authors answered to my concerns. The manuscript is, in my point of view an interesting paper that deserve publication

Reviewer #3 (Remarks to the Author):

Most comments have been addressed in the revised version.

Minor items remain:

- Figure 1, Figure 3: Authors should specify coomassie bands and give size-markers in Supp. Fig 3
- CC 12 is still missing
- Figure S3: please provide the full plots of the images; Comassie-stained version needs size markers; please comment if the prominent bands correspond to MAGL or HSL (they might also be just some proteins of the membrane fractions used for quantification)

Reviewer #4 (Remarks to the Author):

The authors have enhanced the manuscript. However, there are still some issues that need to be improved:

- In fig 3c, the MAGL activity at time 0 is still missing. This information should be necessary to observe the basal levels of both inhibitors to compare with the time curve recovery.
- Figure 2e and 3i are not referenced in the text.
- I strongly encourage to restructure the position of the panels in figure 6. Authors reference the panels in the following order: a, d, b, e, c, f, g.... instead of a, b, c, d...

Response to reviewers

Reviewer #3 (Remarks to the Author):

Most comments have been addressed in the revised version.

Minor items remain:

-Figure 1, Figure 3: Authors should specify coomassie bands and give size-markers in Supp. Fig 3

Reply: done

-CC 12 is still missing

Reply: added.

-Figure S3: please provide the full plots of the images; Coomassie-stained version needs size markers; please comment if the prominent bands correspond to MAGL or HSL (they might also be just some proteins of the membrane fractions used for quantification)

Reply: We have added the uncropped gels with size markers at the end of the supplementary information. The size markers have been added to the coomassie gels in S3. The bands shown in the coomassie do not correspond with MAGL or HSL and were used for quantification.

Reviewer #4 (Remarks to the Author):

The authors have enhanced the manuscript. However, there are still some issues that need to be improved:

- In fig 3c, the MAGL activity at time 0 is still missing. This information should be necessary to observe the basal levels of both inhibitors to compare with the time curve recovery.

Reply: Since there is no time dimension in figure 3c, we assume this comment refers to figure 3b. In this experiment time point 0 will not yield information on the basal levels of MAGL activity. In this experiment the fluorescence of the probe reports the activity. After 0 min of probe labelling there will not be any fluorescent signal.

- Figure 2e and 3i are not referenced in the text.

Reply: corrected.

- I strongly encourage to restructure the position of the panels in figure 6. Authors reference the panels in the following order: a, d, b, e, c, f, g.... instead of a, b, c, d...

Reply: We have rewritten the paragraph in which these panels are referenced so the order is a, b, c, d...